# Structural underpinnings of Ric8A function as a G-protein α-subunit chaperone and guanine-nucleotide exchange factor

Dhiraj Srivastava[1], Lokesh Gakhar [2,3] & Nikolai O. Artemyev [1,4]

Resistance to inhibitors of cholinesterase 8A (Ric8A) is an essential regulator of G protein α-subunits (Gα), acting as a guanine nucleotide exchange factor and a chaperone. We report two crystal structures of Ric8A, one in the apo form and the other in complex with a tagged C-terminal fragment of Gα. These structures reveal two principal domains of Ric8A: an armadillo-fold core and a flexible C-terminal tail. Additionally, they show that the Gα C-terminus binds to a highly-conserved patch on the concave surface of the Ric8A armadillo-domain, with selectivity determinants residing in the Gα sequence. Biochemical analysis shows that the Ric8A C-terminal tail is critical for its stability and function. A model of the Ric8A/Gα complex derived from crosslinking mass spectrometry and molecular dynamics simulations suggests that the Ric8A C-terminal tail helps organize the GTP-binding site of Gα. This study lays the groundwork for understanding Ric8A function at the molecular level.

[1] Department of Molecular Physiology and Biophysics, University of Iowa Carver College of Medicine, Iowa City, IA 52242, USA. [2] Department of Biochemistry, University of Iowa Carver College of Medicine, Iowa City, IA 52242, USA. [3] Protein Crystallography Facility, University of Iowa Carver College of Medicine, Iowa City, IA 52242, USA. [4] Department of Ophthalmology and Visual Sciences, University of Iowa Carver College of Medicine, Iowa City, IA 52242, USA. Correspondence and requests for materials should be addressed to N.O.A. (email: nikolai-artemyev@uiowa.edu)

G protein-coupled receptors (GPCRs) and their cognate heterotrimeric G proteins (Gαβγ) are the key components of a major cellular signaling pathway that is activated when agonist binds to the GPCR, enabling its interaction with Gαβγ and catalyzing the release of GDP from Gα. This enables Gα to bind GTP, triggering dissociation of the signaling species GαGTP and Gβγ, which modulate numerous downstream effectors. Thus, GPCRs serve as guanine nucleotide-exchange factors (GEFs) for G proteins. However, G proteins can also be activated by nonreceptor GEFs. Notable among these are the resistance to inhibitors of cholinesterase 8 (Ric8) proteins, which regulate G-protein biology in species ranging from slime molds to vertebrates and, in contrast to GPCRs, act on monomeric Gα subunits[1–5]. Whereas the genomes of invertebrates encode a single ancestral Ric8 isoform that is capable of interacting with Gα subunits of all types[6,7], their vertebrate counterparts encode two isoforms, each of which regulates a particular subset of Gα subunits: Ric8A (Gα$_{i/t}$, Gα$_q$, and Gα$_{12/13}$) and Ric8B (Gα$_s$)[8–11]. The interaction of Ric8 with GDP-bound Gα stimulates release of GDP, leading to the formation of a stable intermediate complex of Ric8 and nucleotide-free Gα. Once Gα binds GTP it dissociates from Ric8, and thus the nucleotide-exchange cycle on Gα is completed[8,12].

Although the GEF activity of Ric8A and Ric8B was initially thought to account for the ability of these proteins to positively regulate G-protein signaling, mounting evidence suggests that the Ric8 proteins can also serve as ubiquitous chaperones for Gα-subunits[13–15]. The fact that the chaperone function of Ric8 augments G-protein signaling could readily explain many, if not all, biological effects of the protein. Although it remains unclear whether the GEF or chaperone function of Ric8 proteins is predominant with respect to its biological effects, we reasoned that the structure of the Ric8/Gα complex might hold the key to understanding both. This complex underlies the GEF activity of Ric8 and may, in fact, be similar to the folding intermediate that is present during the biosynthesis of Gα[12].

The mechanism underlying the ability of GPCRs to function as GEFs has been extensively investigated. Structures of GPCR/Gαβγ complexes have revealed, in atomic detail, GPCR-induced structural perturbations of Gα that lead to the release of GDP[16–19]. Gα subunits feature two key domains, the Ras-like domain (RD) and the α-helical domain (HD)[20]. The RD binds the nucleotide and interacts with the HD, with the latter serving as a lid over the nucleotide binding site. Agonist-bound GPCR engages Gαβγ at two key sites of Gα: the C-terminal α5 helix and the N-terminal αN-β1 loop. The largest GPCR-induced conformational change in Gα is an outward translation with rotation of the α5 helix that leads to a displacement of the guanine ring binding loop β6-α5 of Gα[16]. Changes associated with the interaction between the receptor and the αN-β1 loop propagate to and disturb the phosphate-binding P-loop (β1-α1). The dual disruption of the guanine-nucleotide binding site is accompanied by weakening of the RD/HD interface and stabilization of conformational states in which RD and HD are dynamically separated, facilitating the escape of GDP[16,21].

In contrast to the wealth of structural information available regarding the interactions between GPCRs and G proteins, data at atomic level have not been available for Ric8 proteins, either alone or in complex with Gα. A lack of sequence similarity between Ric8 isoforms and other proteins has precluded homology modeling of its structure. However, protein-fold recognition algorithms have predicted that Ric8 adopts an armadillo-like fold[22]. The complete lack of structural relatedness between GPCRs and Ric8 suggests that the mechanisms whereby these GEFs activate G proteins are distinct. This notion is supported by the finding that Ric8 cannot interact with heterotrimeric Gαβγ[8].

It had been proposed that Ric8 might interact with the conformationally sensitive switch II region of Gα, which is occluded in the heterotrimer by Gβγ[8,23,24]. This interaction may also explain the selectivity of Ric8 for the GDP-bound state of Gα[8]. Notwithstanding these differences, notable parallels between G protein activation by Ric8A and GPCRs were suggested by some of the available biochemical data. Like GPCRs, Ric8A has been shown to induce separation of RD and HD of Gα, and to do so by perturbing the secondary structure surrounding the guanine-binding site within the RD[25]. A second parallel is that the key Ric8A interaction site of Gα has been localized to the C-terminal α5 helix[26].

To gain mechanistic insights into the function of Ric8 and its interaction with Gα, we solve the crystal structures of both apo Ric8A and Ric8A in complex with the C-terminal fragment of transducin-α (Gα$_t$) attached to a maltose-binding protein (MBP) tag. These structures reveal that Ric8A has two major modules: the core armadillo (ARM)-repeat domain, composed of 8 ARM repeats, and a flexible C-terminal tail. In addition, they show that the concave surface of the Ric8A armadillo domain encompasses a conserved binding site for the C-terminus of Gα. Our biochemical studies demonstrate that the C-terminal tail is important for the stability of the Ric8A protein overall, and key for its GEF function. Modeling of the Ric8A/Gα complex and flexible C-terminal tail of Ric8A provides a foundation for understanding the molecular mechanisms that underlie the stability, GEF function, and chaperone function of Ric8A.

## Results

**Ric8A binds the C-terminus of Gα with high affinity.** The C-terminal 18-mer peptide of Gα$_i$ corresponding to Gα$_t$333–350, was previously shown to bind to Ric8A[26]. To identify C-terminal constructs of Gα that are suitable for crystallization with Ric8A, we tested one in which the 11 C-terminal residues of Gα$_t$ were fused to the B1 domain of Streptococcal protein G (GB1-Gα$_t$340–350) and a second in which the 24 C-terminal residues of Gα$_t$ were fused to MBP (MBP-Gα$_t$327–350). For Ric8A, we used a construct that is truncated at the C-terminus (Ric8A1–492) and that had been reported to retain the full functionality of Ric8A[26]. Elution profiles generated by size-exclusion chromatography (SEC) indicated that both fusion proteins formed complexes with Ric8A1–492 at nearly 1:1 stoichiometry (Supplementary Fig. 1), suggesting that the 11 C-terminal residues of Gα$_t$ are sufficient for a high-affinity (submicromolar $K_d$) interaction.

Next, we evaluated the effects that binding of the 11-mer and 18-mer C-terminal peptides of Gα$_t$ (Gα$_t$340–350 and Gα$_t$333–350, respectively) had on the thermal stability of Ric8A1–492. To this end, we used differential scanning fluorimetry (DSF) (Fig. 1a). These tests revealed that Gα$_t$333–350, but not Gα$_t$340–350, had a strong stabilization effect (~9 °C shift) on Ric8A1–492 (Fig.1a). We infer from the protein stabilization effect that the longer peptide more fully recapitulates the C-terminal interaction of Gα$_t$ with Ric8A. The kinetics and affinity of binding for Gα$_t$333–350 and Ric8A1–492 were quantitated using biolayer interferometry (BLI), with N-terminally biotinylated Gα$_t$333–350 attached to a streptavidin biosensor. The binding and dissociation kinetics for the Ric8A1–492/Gα$_t$333–350 interaction were consistent with a 1:1 binding model, with the average association constant ($k_a$) measured at $1.3 \pm 0.3 \times 10^5$ M$^{-1}$s$^{-1}$ (mean ± SD) and the average dissociation constant ($k_d$) at $0.031 \pm 0.008$ s$^{-1}$, yielding a $K_D = k_d/k_a$ of 0.24 μM (Fig. 1b). The $K_D$ calculated based on steady-state analysis of the interaction between Ric8A1–492 and Gα$_t$333–350 was comparable ($K_D = 0.28 \pm 0.07$ μM, mean ± SD, $n = 4$ (Fig. 1c). Our analysis indicates that the interaction

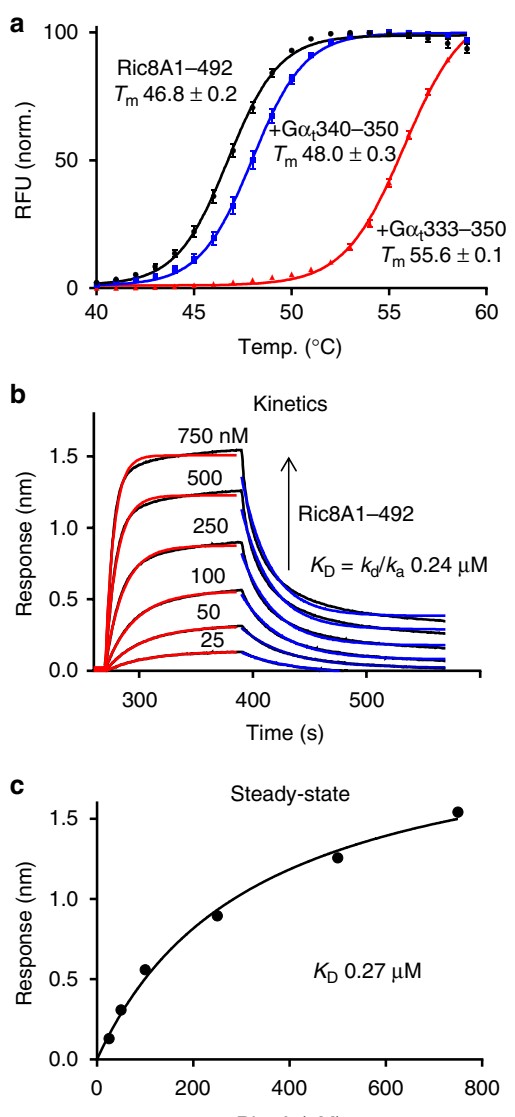

**Fig. 1** Interaction of Ric8A with the C-terminus of Gα. **a** Thermal denaturation, as determined by differential scanning fluorimetry (DSF), of 8.7 μM Ric8A1–492 in the absence (black) or presence of 50 μM Gα$_t$340–350 (blue) or Gα$_t$333–350 (red). RFU – relative fluorescence units. Average normalized curves are shown. The $T_m$ values (°C) are shown as mean ± SD ($n = 3$). **b** Kinetics of association and dissociation for Ric8A1–492 and biotinylated Gα$_t$333–350 coupled to a streptavidin biosensor as determined using BLI. Representative curves are shown. The processed data curves are black and the nonlinear regression fits from the 1:1 binding model are red (association; $k_a = 1.3 ± 0.3 × 10^5 \, M^{-1}s^{-1}$) and blue (dissociation; $k_d = 0.031 ± 0.008 \, s^{-1}$) (mean ± SD). **c** The steady-state binding curve obtained from data in (b); $K_D = 0.27$ μM. For $n = 4$ experiments, $K_D = 0.28 ± 0.07$ μM (mean ± SD). Source data are provided as a Source Data file

between Ric8A and the Gα C-terminus has about 45-fold higher affinity than previously estimated ($K_D$ 12 μM)[26].

**Ric8A is an ARM-repeat protein with a flexible C-terminus.** Crystallization attempts using human Ric8A1–492 in the absence or presence of Gα$_t$333–350 peptide or 24 C-terminal residues of Gα$_t$ fused to MBP (MBP-Gα$_t$327–350) were unsuccessful. However, the use of bovine Ric8A1–492 with an N-terminal His6-tag led to well-diffracting crystals of the protein in complex with

MBP-Gα$_t$327–350 (Supplementary Table 1, Supplementary Fig. 2). Residues at positions 1–426 of Ric8A other than those in the flexible loops (102–106, 202–207, and 292–303) could be modeled clearly based on the electron density (Fig. 2a, b). The electron density for residues 427–492 was missing despite the confirmed presence of unproteolysed Ric8A1–492 in the crystal (Supplementary Fig. 3).

Apo Ric8A1–492 was subsequently crystallized, and the molecular replacement solution for this crystal structure was found using the structure of Ric8A in complex with MBP-Gα$_t$327–350 (Supplementary Table 1, Fig. 2c). As in the case for Ric8A within the complex, only a subset of the Ric8A1–492 residues (Ric8A1–422) could be modeled based on the electron density, suggesting that the C-terminal tail of Ric8A is flexible. Furthermore, comparison of the two structures shows that the binding of MBP-Gα$_t$327–350 to Ric8A does not cause major conformational changes to its core domain, Ric8A1–422 (Fig. 2d). In both structures, ~400 N-terminal residues of Ric8A were packed into a classical armadillo fold consisting of eight ARM repeats (R1–R8; Fig. 2a, b)[27]. R3, 4, 5, 7, and 8 have a canonical ARM repeat structure containing three α-helices, with one short (h1) and running nearly perpendicular to a hairpin formed by two longer antiparallel helices h2 and h3[28]. R1 is a partial ARM repeat that lacks h1, and R2 and R6 appear to be non-canonical ARM repeats, with h1 replaced by a loop. The tandem packing of ARM repeats produces a right-handed ribbon-like superhelix featuring a concave surface formed by the h3 α-helices from R2-R8 (Figs. 2b and 3a, b).

Comparing the Ric8A1–426 structure to those of other proteins in the Protein Data Bank using DALI[29] revealed structural homology with functionally diverse armadillo-fold proteins including β-catenin, importin-α, and the RhoA/RhoC GEF SmgGDS (Supplementary Fig. 4A–C). However, the limited structural similarity (root mean square deviation, RMSD > 3 Å) does not extend beyond ~400 N-terminal residues of Ric8A. Although residues 400–422 of Ric8A form a two-helix hairpin, their orientation and packing differ from those that characterize the armadillo fold (Fig. 2). Moreover, in the structures of apo Ric8A1–492 and Ric8A1–492/MBP-Gα$_t$327–350 conformations of the hairpin are nearly identical despite the different crystal packing interactions that involve residues 400–422 (Supplementary Fig. 5). This suggests that the contacts within the crystal lattice are not responsible for termination of the armadillo fold after R8. Thus, Ric8A contains 8 ARM repeats rather than the 10 that were initially predicted[22]. The apparent structural flexibility of Ric8A427–492 is consistent with its secondary structure prediction (Supplementary Fig. 6).

**The Gα$_t$ C-terminus binds to the concave surface of Ric8A.** The crystal structure of the Ric8A1–492/MBP-Gα$_t$327–350 complex revealed an interface that buries 956 Å² of surface area (total buried area divided by 2)[30]. Residues Gα$_t$333–350 contact the concave surface of Ric8A from R2 to R8 (Fig. 3a, b). The Gα$_t$ C-terminus is not directly involved in crystal contacts. Residues Gα$_t$335–346 form an α-helix that rests over a neutral surface of Ric8A, whereas Gα$_t$347–350 is in an extended conformation and lies over a positively charged surface (Fig. 3a, c). In the final refined structure, all of the side chains of residues Gα$_t$333–350 were modeled with confidence (Fig. 3d, e, Supplementary Fig. 7). The C terminal carboxylate group of Gα$_t$ F350 forms a hydrogen bond and polar interactions with the side-chain amide of N123, as well as the guanidino groups of R71 and R75 of Ric8A (Fig. 3e). The backbone nitrogen atom of F350 also form a hydrogen bond with an oxygen of the side-chain amide of N123. Furthermore, the side-chain of F350 is involved in the T-shaped π–π stacking

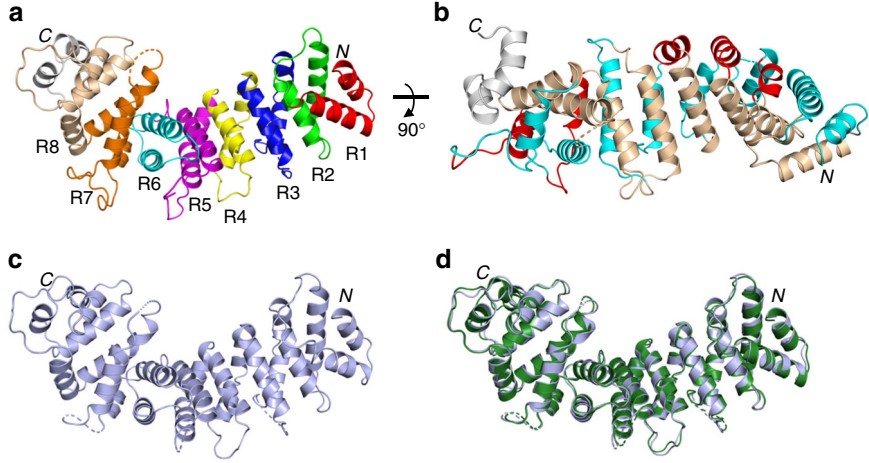

**Fig. 2** Structure of the Ric8A armadillo core. Structure of Ric8A1–426 based on the Ric8A1–492/MBP-Gα$_t$327–350 complex. **a** ARM repeats R1-8 are shown in different colors, as indicated. **b** α-helices h1, h2, and h3 comprising each ARM-repeat are shown in red, cyan, and wheat, respectively. The two-helix hairpin of Ric8A (residues 400–426) is shown in gray. **c** Structure of apo Ric8A1–492 in which only residues 1–422 were resolved. **d** Overlay of the structures in (**a**) and (**c**)

interaction with F163 of Ric8A, which is held in place by the T-shaped $\pi$–$\pi$ stacking interaction with the neighboring F162 (Fig. 3e). Another hydrogen bond is formed between the side chains of Gα$_t$ N343 and Ric8A H273. Besides these hydrogen bonds and polar interactions, most of the interactions between Ric8A and the Gα$_t$ C-terminus are hydrophobic or side-chain packing interactions. For instance, I339 and I340 of Gα$_t$ occupy a hydrophobic patch formed by F228 and F232 of Ric8A (Fig. 3e). By binding across the concave surface of Ric8A, the Gα$_t$ C-terminus could act as a scaffolding element that bridges multiple ARMs. Such a scaffolding interaction could underlie the increase in the thermal stability of Ric8A1–492 in the presence of the Gα$_t$333–350 peptide.

To probe the protein interface observed in the crystal structure we tested the ability of Gα$_t$333–350 to bind to Ric8A1–492 proteins that were mutated to disrupt this interaction (Fig. 3f, Supplementary Fig. 8). The substitutions, R75M and N123E, were predicted to interfere with the interaction network of Gα$_t$ F350. These mutations had the most severe consequences, as they ablated the binding of Ric8A1–492 to Gα$_t$333–350 in the BLI assay. The substitutions, F169R and A173F, introduced steric hindrance, which also severely impaired the interaction (Fig. 3f). Similar results were obtained using the full-length Avi-tagged Gα$_t$ in the BLI assay. Ric8A1–492 bound to Gα$_t$ with high affinity ($K_D$ 24 ± 2 nM, mean ± SD, $n$ = 3; Fig. 3g, Supplementary Fig. 9A). The F169R and A173F mutations severely reduced affinity of Ric8A1–492 for Gα$_t$, whereas no Gα$_t$ binding was detected using the N123E mutant (Fig. 3g, h; Supplementary Fig. 9B). In addition, the disruption of the interaction of Ric8A with the full-length Gα$_t$ was observed using the Ric8A R75M mutant in SEC co-migration experiments (Supplementary Fig. 10). Thus, mutational analysis validated the Gα$_t$ binding site on Ric8A identified in our crystal structure.

To assess the conformation of the C-terminal region of Gα in the absence of intramolecular interactions we solved the crystal structure of MBP-Gα$_t$327–350 alone (Supplementary Table 1). Although the Gα$_t$327–348 sequence could be traced in electron density, it largely lacked secondary structure and the conformation differed markedly from that in the complex containing Ric8A (Supplementary Fig. 11). Thus, binding of Gα$_t$327–350 by Ric8A induces the formation of an α-helix by Gα$_t$335–346 and/or stabilizes such a structure.

**The Gα C-termini contribute to the Ric8 isoform selectivity**. The interface between Ric8A and the Gα$_t$ C-terminus involves both residues that are conserved across all families of Gα subunit (such as Gα$_t$ I339, L344, and L349) and residues that are conserved only in the Gα$_{i/t}$ family (V335, T336, I340, C347, and F350)[31]. In contrast, nearly all of the Ric8 residues that interact with Gα are strongly conserved across the Ric8A and Ric8B isoforms, as well as the ancestral Ric8 proteins. (Fig. 4, Supplementary Data 1)[32]. The conservation analysis, therefore, suggests that family-specific residues of Gα, but not of Ric8 proteins, are the determinants of selective coupling between these proteins. We tested this notion, using BLI to examine the binding of Ric8A to a peptide comprised of the C-terminal 18 residues of Gα$_s$ (Gα$_s$363–380). Notably, Ric8A was capable of binding to Gα$_s$363–380, albeit with a ~18-fold lower affinity than to Gα$_t$333–350 (Supplementary Fig. 12a–c).

Examination of the Ric8A/Gα$_t$327–350 interface in the crystal structure suggested that many of the contacts would be maintained if Gα$_t$ residues were substituted for their Gα$_s$ counterparts. However, we considered that the four C-terminal Gα$_t$ residues, CGLF, may provide a better fit for the extensive packing interactions with Ric8A. We therefore tested a chimeric C-terminal peptide of Gα$_s$ ending in CGLF (peptide C1) (Supplementary Fig. 12a, d, e). Although the binding of C1 to Ric8A was not enhanced, a peptide containing two additional substitutions, Gα$_s$Q370I (I340 in Gα$_t$) and Gα$_s$H373N (N343 in Gα$_t$) (peptide C2), resulted in potent Ric8A binding, similar to that for Gα$_t$333–350 (Supplementary Fig. 12a, f, g). Moreover, the double I340Q/N343H mutation of Gα$_t$ decreased its affinity for Ric8A1–492 by about 100-fold ($K_D$ = 2.3 ± 0.4 μM, mean ± SD, $n$ = 3; Supplementary Fig. 13). These findings suggest that residues at positions I340 and N343 of Gα$_t$ are important for the selectivity of interactions between Gα$_{i/t}$ proteins and Ric8A. Regions of Gα other than the C-termini may also be involved in the selective interaction with Ric8 isoforms given that the Gα$_s$ C-terminal peptide interacted with Ric8A to some extent.

The strong conservation of the Gα C-terminal binding surfaces in Ric8A and Ric8B raised the possibility that Ric8B interacts with the C-termini of Gα$_{i/t}$-like Gα subunits. However, no interaction between Gα$_t$333–350 and Ric8B was detected, even when a high concentration of Ric8B (20 μM) was used. Hence, the

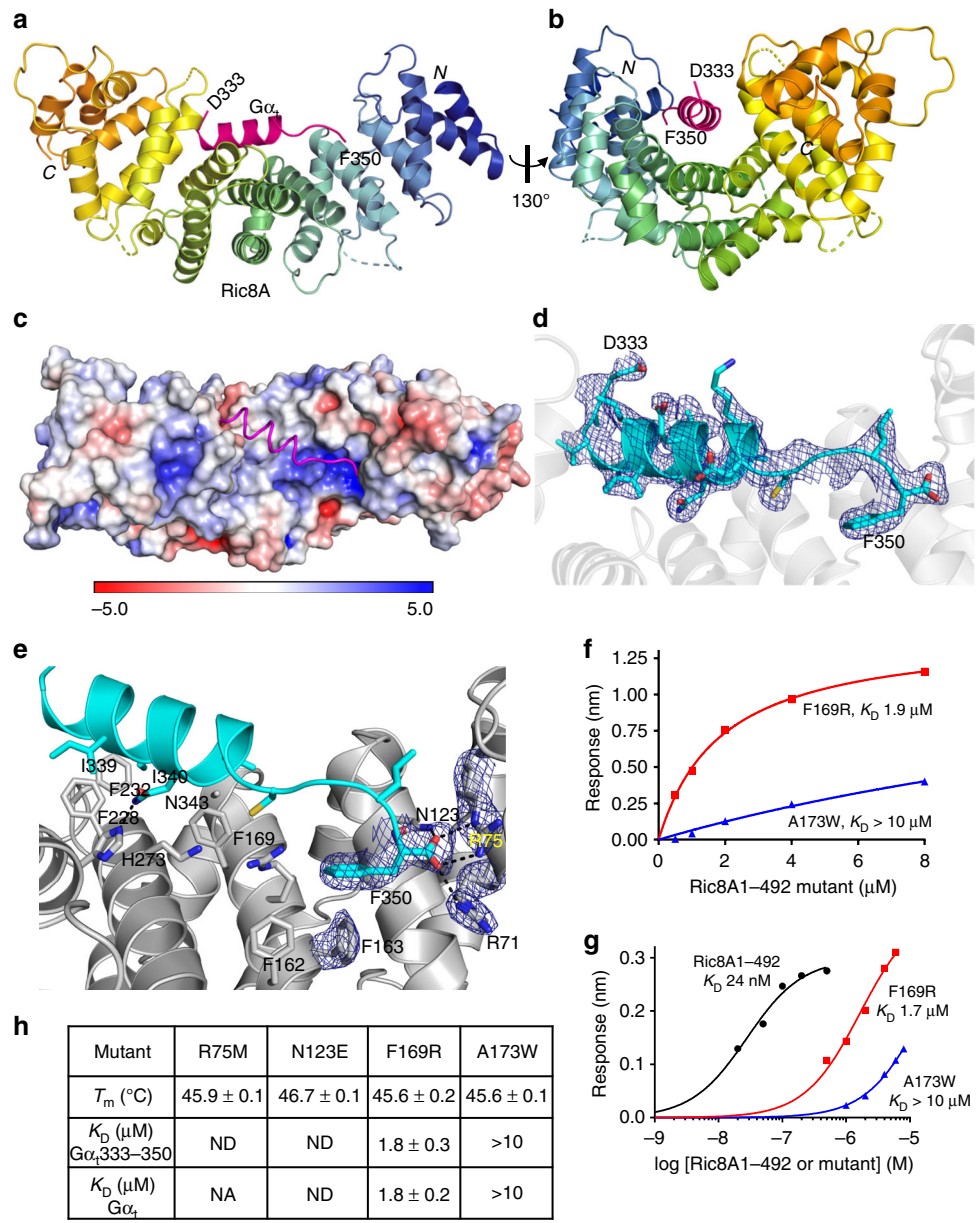

**Fig. 3** Interface of Ric8A with the C-terminus of Gα$_{t1}$. **a** View of Gα$_t$333–350 (magenta) bound to the concave surface of Ric8A (rainbow), based on the structure of the Ric8A1–492/MBP-Gα$_t$327–350 complex. **b** View highlighting curvature of the concave binding surface of Ric8A. **c** Electrostatic surface representation of Ric8A with the bound Gα$_t$333–350 (magenta tube), showing helical residues Gα$_t$335–346 interacting with the neutral surface and the C-terminal-most residues contacting the positively charged patch (units $K_bT/e_c$). **d** An omit map (*Fo-Fc*) for Gα$_t$333–350 is contoured at 2.5σ and shown for 1.6 Å around the ligand. **e** Close-up of the interface between Ric8A (gray) and the Gα$_t$ C-terminus (cyan). Omit maps (*Fo-Fc*) for Gα$_t$ F350 and its critical Ric8A contact residues R71, R75, N123, and F163 are contoured at 2.5σ and shown for 1.6 Å around the side chains. **f, g** Properties of the mutations that alter Ric8A/Gα$_t$333–350 interface. Representative steady-state binding curves from BLI assays measuring the binding of Ric8A1–492 mutants F169R and A173F to biotinylated Gα$_t$333–350 (**f**) or Ric8A1–492 and its F169R and A173F mutants to Avi-tagged Gα$_t$ (**g**) coupled to a streptavidin biosensor. **h** Table showing thermostability (DSF assay) and Gα$_t$333–350 and the full-length Gα$_t$ steady-state binding affinities of Ric8A1–492 mutants calculated from the BLI assays (mean ± SD, *n* = 3). ND – not detectable. NA – not available. Source data are provided as a Source Data file. Stereo images of 3d and 3e are available in Supplementary Fig. 7

conformation of the Ric8B site that binds the Gα C-terminus may differ from the one on Ric8A.

**The Ric8A stability is regulated by its proximal C-terminus.** The structure of the Ric8A1–492/MBP-Gα$_t$327–350 complex indicates that the flexible C-terminal region Ric8A427–492 is not directly involved in binding the Gα C-terminus. However, this region is known to contain a potential Gα binding site[33]. To determine the role of Ric8A427–492 in the function of this

protein, we first used BLI to assess an interaction between Ric8A1–426 (lacks the C-terminal region) and Gα$_t$333–350. Unexpectedly, the Gα$_t$ C-terminal peptide did not bind to Ric8A1–426 (Supplementary Fig. 14A). To probe the mechanism whereby Ric8A427–492 contributes to the interaction with Gα$_t$333–350, we examined the thermal stability of Ric8A1–426 by DSF. This analysis revealed that Ric8A1–426 ($T_m$ = 36.5 ± 1.1 °C, mean ± SD, *n* = 3) was much less stable than Ric8A1–492 (Supplementary Fig. 14B). Thus, the C-terminal tail of Ric8A seems to

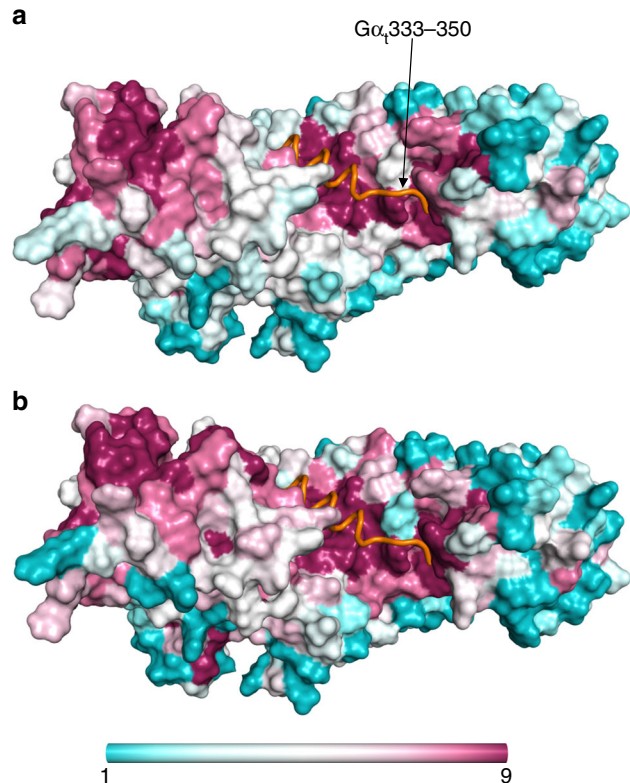

Gα_t333–350

**Fig. 4** Binding surface for the Gα_t C-terminus is highly conserved in Ric8 isoforms. Surface representations for Ric8A1–426 colored by residue conservations scores derived from ConSurf analysis of a sample of 250 Ric8 homologs (**a**) and 117 Ric8A orthologs (Ric8A subtree) (**b**). The sequences for the analysis with maximal identity of 97% and minimal identity of 30% were collected from the UniProt database. Color scale: 9 – magenta – conserved, 1 – cyan –variable. Gα_t333–350 is shown as an orange tube

be engaged in intramolecular interactions with the core domain, Ric8A1–426, both stabilizing it and allosterically promoting its binding to the Gα C-terminus.

To map these intramolecular interactions, we generated the Ric8A construct, Ric8A1–452, in which less of the C-terminal region is missing. Both the thermal stability of Ric8A1–452 and its ability to bind to and be stabilized by Gα_t333–350 were comparable to those of Ric8A1–492, indicating that proximal C-terminal residues 427–452 are essential (Fig. 5a, Supplementary Fig. 15). This segment is conserved in the Ric8A and Ric8B isoforms, and it is also highly acidic.

To identify intramolecular interaction sites involving the C-terminal region of Ric8A1–492 we utilized the crosslinking mass spectrometry (XL-MS) approach, with disuccinimidyl suberate (DSS) serving as the crosslinking agent. Nine crosslinked pairs were identified using the Protein Prospector software[34]. Among these pairs, four involved K449 in the flexible C-terminal region (Supplementary Data 2, Fig. 5e). In particular, K449 was crosslinked to K349, K352, K375, or K408. Residues K349, K352, and K408 together with R345, R348, and R405 form a large and highly conserved positively charged patch on the outer surface of the C-terminal portion of the Ric8A core domain (Fig. 5c, d, Supplementary Fig. 5D). Thus, the crosslinking analysis is consistent with the notion that electrostatic interactions between the negatively charged proximal C-terminal segment Ric8A427–452 and the positively charged surface on the Ric8A core domain play an important role in stabilizing Ric8A.

**A model of Ric8A C-terminus accounts for its stabilization.** We employed the FloppyTail application within the Rosetta framework to model the structure of the C-terminal region of Ric8A in its apo form and in complex with Gα[35]. First, we modeled the structure of the proximal portion of the Ric8A C-terminal tail. A total of 4900 models of Ric8A1–452 were generated, from which 212 models were selected based on the energy scores, crosslinking constraints and agreement ($\chi^2$ values) between theoretical SAXS profiles for models and the experimental SAXS profile of Ric8A1–452 (see Methods, Fig. 5b, and Supplementary Table 2, Supplementary Fig. 16). Clustering of 212 models yielded two main clusters I and II that were comprised of 50 and 20 models, respectively (Supplementary Fig. 17)[36]. In all of the highest scoring models, the basic patch in the core Ric8A1–426 and the acidic stretch Ric8A436–444 (EDEDTDTDE) were in close proximity and shared contacts. Cluster I included 5 of the top 10 energy score models. Cluster II models were rejected because they predicted that the orientation of the C-terminal end was away from the Gα-binding site, making the reported interaction of the Ric8A C-terminal tail with Gα highly improbable (Supplementary Fig. 17)[33]. Thus, the highest scoring model of Ric8A1–452 from cluster I was selected for further analysis and modeling (Fig. 5c, d).

Molecular dynamics (MD) simulations of the Ric8A1–452 model were performed to determine if and how it might account for the higher thermal stability of Ric8A1–452 vs. Ric8A1–426. MD simulations of the Ric8A1–426 structure and the Ric8A1–452 model revealed that the root mean square fluctuation (RMSF) values for residues in the mobile regions are significantly reduced in the Ric8A1–452 model, indicating that the protein is stabilized (Fig. 5f, Supplementary Fig. 18).

The conformation of the distal portion of the C-terminal tail (Ric8A453–492) was assessed using the experimental SAXS profile of Ric8A1–492 and a conformational sampling analysis using the BILBOMD server[37] (Supplementary Figs. 16 and 19). The best-scoring single state and three-state models of apo Ric8A1–492 suggest that the distal portion of the C-terminal tail assumes extended conformations and does not contact the core of the protein (Supplementary Fig. 19).

**Mechanistic insights from modeling of the Ric8A/Gα complex.** Our initial model of the Ric8A/Gα complex was generated by superimposing the α-helix (residues Gα_t335–340) from Gα_tGDP (PDB 1TAG) and the Ric8A1–492/MBP-Gα_t327–350 structure (PDB 6N85). This overlay revealed extensive overlap and clashes between Gα and the C-terminal half of Ric8A, suggesting that major conformational changes are needed to accommodate this interaction (Supplementary Fig. 20A). A similar overlay using the Gα conformation from the GPCR/Gα structure (PDB 3SN6) significantly reduced these clashes, and thus was selected for further modeling (Supplementary Fig. 20B)[16]. To simplify the modeling, we took advantage of the functionality of reduced miniGα constructs, which lack the HD domain[38,39]. A homology model of miniGα_i lacking the αN-helix (ΔN25-miniGα_i) was generated using a structure of the GPCR-bound miniGα_s[38]. MD simulations of the of Ric8A1–452 model (Supplementary Fig. 18) and ΔN25-miniGα_i (Supplementary Fig. 21) allowed us to select conformations of these molecules that minimized the steric clashes. To simulate the forces that acted on Ric8A upon binding of ΔN25-miniGα_i, we conducted steered MD (SMD) simulations (Supplementary Fig. 22). This SMD simulation led to a slight twisting of the C-terminal module and produced an open conformation of Ric8A1–452 with reduced concave curvature (Supplementary Fig. 20C). Ric8A1–452 in open conformation would not clash with ΔN25-miniGα_i upon superimposition of the C-terminal helical segments (Supplementary Fig. 20D).

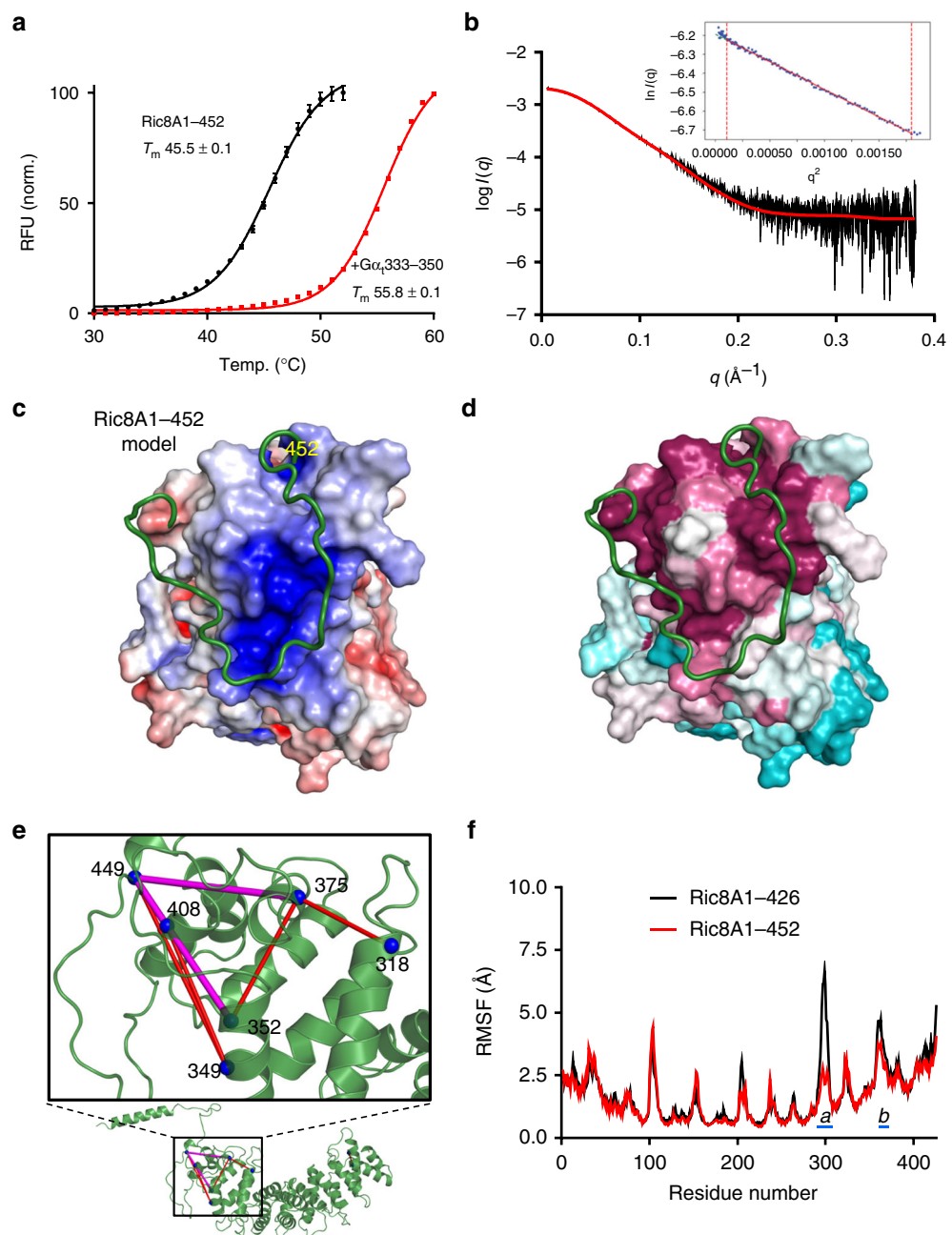

**Fig. 5** Proximal portion of the Ric8A C-terminal tail binds and stabilizes the armadillo core domain. **a** Denaturation of 9.1 μM Ric8A1–452, as determined by DSF, in the absence (black) or presence of 50 μM Gαt333–350 (red). RFU – relative fluorescence units. The $T_m$ values (°C) are shown as mean ± SD. **b** Experimental SAXS data for Ric8A1–452 (black). Theoretical SAXS profile calculated for the Ric8A1–452 model fits the SAXS data with a $\chi^2$ value of 1.82. The Guinier plot for the low q region ($q \cdot R_g < 1.3$) is shown in the inset. **c**, **d**. The best-scoring FloppyTail model of Ric8A1–452 from cluster I. The proximal C-terminal region Ric8A423–452, shown as a ribbon, the core domain Ric8A1–422 shown as electrostatic surface representation (units $K_bT/e_c$) (**c**) or as a conservation surface (based on ConSurf analysis Ric8 homologs (**d**); the color scale represents conservation scores, 9 – magenta – conserved, 1 – cyan - variable). **e**. Close-up of the model of apo Ric8A1–492 showing the region with the highest concentration of intramolecular DSS-crosslinks. Eight of the total nine crosslinks (blue) were identified in the C-terminal half of Ric8A. Crosslink K352/K449 is obscured by crosslinks K352/K408 and K408/K449. All the crosslinks satisfy the distance threshold of 30 Å[74]. Crosslinks K352/K449 and K408/K449 used in FloppyTail modeling of Ric8A1–452 and crosslink K375/K449 used in model selection are colored in magenta; all other crosslinks are in red. **f** RMSF plots averaged from four MD simulations (Supplementary Fig. 18) for each Ric8A1–426 (black) and the model of Ric8A1–452 (red). The flexible regions **a** and **b** of Ric8A significantly stabilized by the proximal part of its C-terminal tail are indicated with a blue line. The peak average RMSF values (Å) for **a** (6.7 ± 1.4 vs. 2.1 ± 0.4 (mean ± SE, $n = 4$), unpaired $t$-test *$P = 0.02$, residue 299) and **b** (4.7 ± 0.2 vs. 3.7 ± 0.1, unpaired $t$-test **$P = 0.004$, residue 363) are statistically different. Source data are provided as a Source Data file

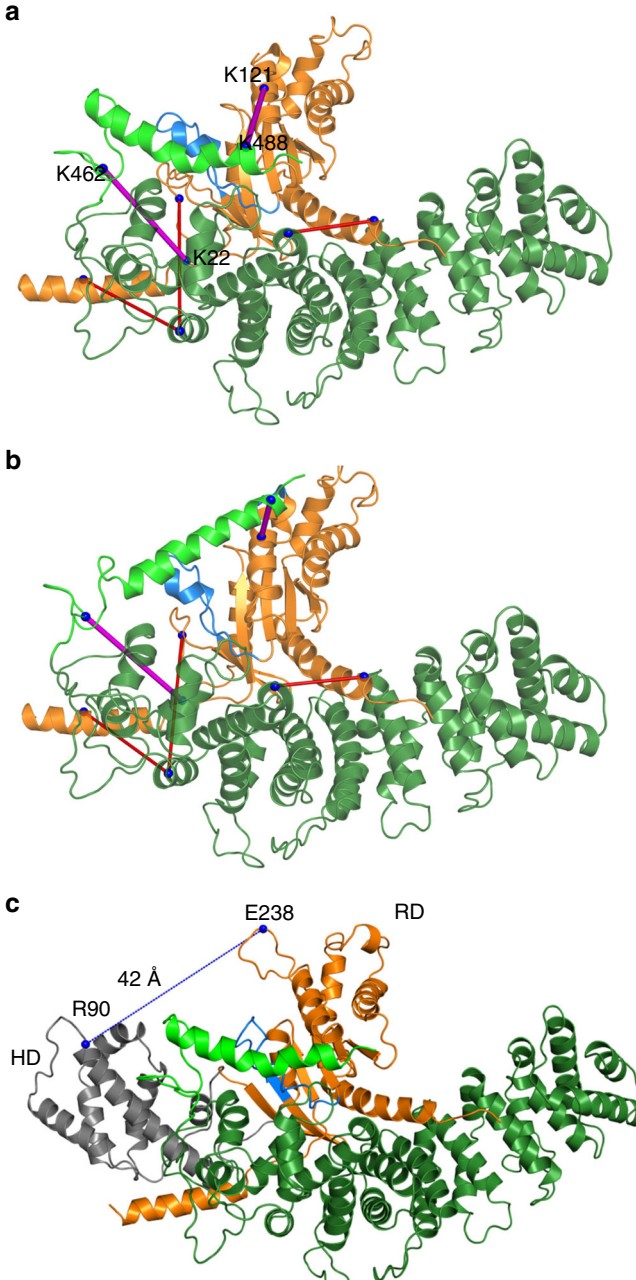

**Fig. 6** Models of the Ric8A/Gα complex. **a** FloppyTail model 1 from cluster I of Ric8A1–492/ΔN19-miniGαi models with the addition of 19 N-terminal residues of miniGαi. Ric8A is shown in green and ΔN19-miniGαi is shown in orange with the switch II region shown in blue. Distal C-terminal tail residues 455–470 extend beyond the C-terminal end of the Ric8A core domain, and the Ric8A 471–490 helix (light green) is situated along the switch II region (blue) and the α3-β5 loop. Five intermolecular crosslinks between Ric8A1–492 and miniGαi were identified by XL-MS analysis and Protein Prospector[61]. All of the crosslinks satisfy the distance threshold of 30 Å[74]. The crosslink pairs that involve the distal portion of the Ric8A C-terminal tail and that were used in modeling (K488/miniGαi-K122 and K462/miniGαi-K21) are shown in magenta; all other crosslinks are in red. **b** FloppyTail model 2 from cluster II of Ric8A1–492/ΔN19-miniGαi models. The Ric8A 471–490 helix (isolated light green helix) hangs over the switch II region (blue) and points towards the switch III region. The crosslinks are shown as in (**a**). **c** Model of the Ric8A1–492/Gαi complex with a 42-Å separation of the α-helical domain (HD, gray) and Ras-like domain (RD, orange) of Gαt

Our biochemical assays confirmed an earlier report that deletion of the distal portion of the Ric8A C-terminal tail severely diminishes the GEF activity of this protein (Supplementary Fig. 23)[26]. This and the identification of a potential Gα-binding site at residues Ric8A455–470[33] suggest that this portion of the C-terminal tail is essential for its functional interaction with Gα. To characterize this interaction, we conducted XL-MS on purified complexes of Ric8A1–492 with miniGαi (Supplementary Data 3) and Gαt (Supplementary Data 4, Supplementary Fig. 24). Gαt and miniGαi shared intermolecular crosslinks with the core domain of Ric8A (Fig. 6a), all of which were consistent with the Ric8A1–452/ΔN25-miniGαi model (Supplementary Fig. 20D). Importantly, the crosslinking of Ric8A1–492 with Gαt and miniGαi identified one and two, respectively, crosslinked pairs involving the distal portion of the Ric8A C-terminal tail: Ric8A-K488/Gαt-K244, Ric8A-K488/miniGαi-K122 (equivalent to Gαt-K244), and K462/miniGαi-K21 (equivalent to Gαt-K25) (Fig. 6a, Supplementary Data 3 and 4). The latter two crosslinked pairs were used to constrain the FloppyTail modeling of Ric8A453–492. To allow the K462/miniGαi-K21 crosslink constraint to be used in the modeling, the N-terminus of the ΔN25-miniGαi was extended to ΔN19-miniGαi.

In total, 5548 FloppyTail models were generated, clustered and sorted according to their energy scores (Supplementary Fig. 16). The two main clusters obtained using an RMSD value of 2.5 Å contained 14% (cluster I) and 11 % (cluster II) of models (Supplementary Fig. 25). The mean values for the energy score were −1652 for cluster I and −1657 for cluster II. In the majority of models, the distal C-terminal tail residues 455–470 rise above the C-terminal end of the Ric8A core domain, in proximity to the switch II region of Gα. In cluster I models, the Ric8A 471–490 α-helix is situated along the switch II region and the α3-β5 loop, whereas in cluster II models it hangs over the switch II region, pointing towards the switch III region. (Supplementary Fig. 25, Fig. 6a, b). Both the top cluster I and II models, exemplified by model 1 (energy score −1693) and model 2 (energy score −1690), respectively, were ranked high with respect to energy scores (Fig. 6a, b). Both models are plausible and do not differ in critical respects.

A model of the Ric8A1–492/Gαi complex was generated based on model 1 of the Ric8A1–492/miniGαi complex and the structure of the rhodopsin/Gi complex (Fig. 6c)[40]. The distance distribution 20–48 Å for the spin pair at residues 90 and 238 in Gαi has been experimentally determined for the Ric8A/Gαi complex as a measure of the degree of HD/RD separation[25]. In the model of the Ric8A1–492/Gαi complex the distance between Cα atoms of Gαi R90 and E238 is 42 Å. Furthermore, the HD in the model is free to sample all conformations with the reported distance distribution, and it does not appear to come into proximity with Ric8A.

## Discussion

In this study, we solved the structures of apo Ric8A alone and in complex with a fusion protein containing the Gαt C-terminus and MBP. These structures revealed that the core domain of Ric8A has an armadillo-type fold comprised of 8 ARM repeats and that the highly conserved area on the concave surface of the Ric8A superhelical armadillo array serves as a binding site for the C-termini of Gα proteins. Additional binding experiments revealed that the selectivity of the Ric8A/Gα interaction is determined, in part, by a few class-specific residues of Gα, particularly those at positions corresponding to I340 and N343 of Gαt. Armadillo proteins that are structurally related to Ric8A, including SmgGDS β-catenin, and importin-α, commonly bind their partners at the concave surface[41–43]. However, the nature of the ligand-binding

interactions of Ric8A differs from those of other ARM-repeat proteins (Supplementary Fig. 26).

Solving the structure of Ric8A1–426 in complex with the C-terminus of $G\alpha_t$ allowed us to model the entire Ric8A/G$\alpha$ complex because we could superimpose the $G\alpha_t$ helical fragment from the complex with the $\alpha$5 $\alpha$-helix of the G$\alpha$ subunit in the GDP, GTP, and GPCR-bound conformations. The significant reduction in clashes between G$\alpha$ and Ric8A when the GPCR-bound conformation of G$\alpha$ was used was intriguing. It suggested that, on binding to Ric8A and GPCR, the initial conformational changes of the G$\alpha$ subunit are similar and involve $\alpha$5 translation along, and rotation around, the helical axis, i.e., movement away from the GDP-binding site[16]. This distal conformation of $\alpha$5 is rare in G$\alpha$GDP, but it is dominant in complexes between agonist-bound GPCRs and G$\alpha\beta\gamma$[21]. Our finding that Ric8A is capable of high-affinity binding to a region as small as the 11-C-terminal residues of $G\alpha_t$ suggests that Ric8A may also shift the equilibrium in G$\alpha$GDP to the distal $\alpha$5 conformation. Furthermore, the Ric8A-bound $\alpha$5-helix of G$\alpha$ displays an extension similar to that observed during the G$\alpha$ C-terminal disorder-to-order transition when G$\alpha\beta\gamma$ binds to GPCRs (Supplementary Fig. 11B)[16,44].

Modeling of the GPCR-bound conformation of G$\alpha$ in the complex with Ric8A did not prevent all clashes; further conformational changes in Ric8A were needed for this. SMD simulations indicated that Ric8A also assumes an open conformation to accommodate G$\alpha$. The resulting model of the Ric8A/G$\alpha$ complex is consistent with our XL-MS analysis, and it indicates that in addition to binding the $\alpha$5-helix of G$\alpha$, the Ric8A core domain extensively interacts with the $\alpha$N-$\beta$1 loop and the switch II region of G$\alpha$. These interactions disrupt the G$\alpha$ GDP-binding site and separate of the HD and RD domains[25]. Although the $\alpha$N-$\beta$1 loop region is a part of the interface between G$\alpha\beta\gamma$ and GPCRs, the interaction of the switch II region of G$\alpha$ would be unique for the Ric8A/G$\alpha$ complex. Furthermore, the modeled Ric8A/G$\alpha$ interface is incompatible with binding of Ric8A to heterotrimeric G proteins.

Our studies have revealed critical roles for the seemingly unstructured C-terminal region of Ric8A (residues 427–492). The proximal portion of the Ric8A C-terminal tail (residues 427–452) was found to be important for stability of the protein. Further our XL-MS studies and molecular modeling suggest that the negatively charged stretch within Ric8A427–452 interacts with the conserved positively charged region of the Ric8A armadillo core. MD simulations confirmed that this interaction can markedly stabilize the most flexible regions of Ric8A. The finding that, in the crystal structure, the G$\alpha$ C-terminus is bound to the Ric8A armadillo core despite the fact that the C-terminal tail of Ric8A lacks electron density appears to be serendipitous. In the crystal lattice, the Ric8A core is stabilized by contacts between its positively charged region and the negatively charged domain of MBP (Supplementary Fig. 5), which may displace the C-terminal tail. Phosphorylation of S435 and T440 of Ric8A by the protein kinase CK2 has been recently shown to enhance Ric8A binding to G$\alpha$ subunits along with increasing its GEF and chaperone activities[45]. Our results suggest that instead of directly a being part of the G$\alpha$ binding site, these phosphorylation sites augment the stabilizing intramolecular electrostatic interactions in Ric8A and thereby potentiate Ric8A activities.

Our analysis supports the idea that the C-terminal tail of Ric8A, Ric8A453–492, plays a central role in the GEF activity of the protein. It organizes or stabilizes regions of G$\alpha$ that are disordered upon GDP dissociation, thereby enabling G$\alpha$ to bind GTP. As has been shown biochemically, truncated Ric8A1–453 retains substantial capacity to dissociate GDP from G$\alpha$, yet Ric8A1–453-bound G$\alpha$ is unable to bind GTP[26]. Furthermore, compelling evidence points to residues Ric8A455–470 being a

binding site for G$\alpha$, and to the switch II region of G$\alpha$ being an important binding site for Ric8A[8,33]. Our models of the position and conformation of the distal portion of the Ric8A C-terminal tail are consistent with existing biochemical evidence. In apo Ric8A, this part of the tail appears to extend away from the protein core. In contrast, in the Ric8A/G$\alpha$ complex it reaches towards G$\alpha$ and engages the switch II and, possibly, switch III regions, which are likely to be essential for the ability of G$\alpha$ to bind GTP. Once GTP is bound, the switch regions change conformation, preventing G$\alpha$ from binding to the distal portion of the Ric8A tail, and the interaction between G$\alpha$ and the armadillo core is diminished by retraction of the $\alpha$5 $\alpha$-helix, resulting in dissociation of G$\alpha$GTP from Ric8A.

Overall, the structure-based model of the Ric8A/G$\alpha$ complex presented here suggests parallels, as well as differences, in the mechanisms underlying the GEF activities of GPCRs and Ric8A. Specifically, GPCRs and Ric8A engage some of the same regions of G$\alpha$, the C-terminus and the $\alpha$N-$\beta$1 region[16]. Yet the Ric8A/G$\alpha$ interface appears to be more extensive and Ric8A seems to induce more profound structural perturbations of G$\alpha$ near the switch II region and guanine-nucleotide binding site[33]. Thus, Ric8A appears to use the distal portion of its tail to organize, or form de novo, the guanine-nucleotide binding site on G$\alpha$, as this would be required for its chaperone activity and GTP-binding.

## Methods

**Protein expression and purification.** For crystallography and biochemical studies, sequences encoding bovine Ric8A1–492, Ric8A1–452, and Ric8A1–426 were amplified from a Ric8A cDNA clone obtained from Dharmacon (accession number NM_001015627.2) and cloned into the NdeI-XhoI sites of the pET15b vector (Novagen). Ric8A1–492 mutations were introduced using the QuikChange protocol for site-directed mutagenesis. For Small Angle X-ray Scattering (SAXS) experiments, Ric8A1–492 and Ric8A1–452 were cloned into a modified pRSF Duet vector (Novagen), with a 34 amino-acid linker introduced between the 6xHis-tag and TEV protease cleavage site, and a 3 amino-acid linker introduced between the TEV protease cleavage site and Ric8A. Sequences coding for G$\alpha_s$, G$\alpha_i$, and chimeric G$\alpha_t$[46,47] were cloned into the NcoI-XhoI sites of a modified pET21a vector containing an N-terminal 6xHis-tag and a TEV protease cleavage site. For the cloning of minimal G$\alpha_i$ (miniG$_i$), the 8 N-terminal residues of G$\alpha_{i1}$ were deleted, a helical domain corresponding to residues 61–177 was replaced with the Glu-Glu epitope tag (EYMPME) linker, residues 231–237 from the switch III region were deleted, and six amino-acid substitutions (G41D/E41N/G219D/T221A/A228D/P290Q) were introduced using overlap extension PCR[39,48,49]. This miniG$_i$ was cloned into the NcoI-XhoI sites of a modified pET21a vector. All PCR primers used in this study are listed in Supplementary Table 3.

For crystallography, we generated a surface entropy reduction mutant of Ric8A1–492, [460]EEK[462]→AAA[50]. For crystallography and biochemical studies, we generated a fusion protein of G$\alpha_t$340–350 with the B1 domain of Streptococcal protein G (GB1-G$\alpha_t$340–350) and a fusion protein of G$\alpha_t$327–350 with maltose-binding protein (MBP). Sequence G$\alpha_t$340–350 was chosen because the corresponding G$\alpha_t$ peptide potently binds to activated rhodopsin, and the 11 C-terminal residues of G$\alpha_{i/t}$ form the most important interface with rhodopsin[40,51]. The sequence coding for G$\alpha_t$340–350 was cloned into BamHI-HindIII sites of a modified pQE30 vector containing the GB1 tag. A longer sequence G$\alpha_t$327–350 was used in the MBP-fusion protein to allow for a linker, such that potential crystal packing interactions would not interfere with the known interaction involving the C-terminal 18-mer peptide of G$\alpha$[26]. MBP was modified by introducing multiple mutations that improve its ability to crystalize[52,53]. E. coli MBP28–394 with amino-acid substitutions D110A/K111A/E200A/N201A/A243H/K247H/K267A was attached to G$\alpha_t$327–350 via an AAAH linker using overlap extension PCR. The construct was cloned into the NdeI-XhoI sites of the pET15b vector.

All constructs were transformed into E. coli strain BL21(DE3) (Novagen). Cells expressing Ric8A1–492 and Ric8A1–452 were grown to $OD_{600} = 0.6$ in Terrific Broth (TB) medium at 37 °C and induced with 0.5 mM IPTG at 22 °C overnight. Cells expressing Ric8A1–426 were grown to $OD_{600} = 0.6$ in TB medium at 37 °C and induced with 0.5 mM IPTG at 22 °C for 2 h. For the expression of all full-length G$\alpha$ constructs, cells were grown to $OD_{600} = 0.6$ in 2TY medium at 37 °C and induced with 50 µM IPTG at 17 °C overnight. Cells expressing miniG$\alpha_i$ were grown to $OD_{600} = 0.6$ in LB medium supplemented with 2 mM MgSO4 at 37 °C and induced with 50 µM IPTG at 17 °C overnight. Cells expressing MBP-G$\alpha_t$327–350 were grown to $OD_{600} = 0.6$ in LB medium at 37 °C to $OD_{600} = 0.6$ and induced with 0.5 mM IPTG at 37 °C overnight.

Cells expressing 6xHis-tagged Ric8A1–492, Ric8A1–452, Ric8A1–426, G$\alpha_t$, G$\alpha_s$, G$\alpha_i$ or MBP-G$\alpha_t$327–350 were resuspended in buffer N1 (50 mM HEPES, 300 mM NaCl, 5% glycerol, pH 8.0) supplemented with a Complete™, Mini, EDTA-free

Protease Inhibitor Cocktail tablet (Roche) and 2 mM PMSF. For Gα constructs, cell suspensions were also supplemented with 10 mM MgCl₂ and 50 μM GDP. Cells were lysed by sonication, cell debris was cleared by centrifugation and supernatant was loaded onto His-bind resin (EMD Millipore) charged with Ni⁺⁺. Resin was washed with 5-column volumes of resuspension buffer followed by buffer N1 containing 30 mM imidazole. Proteins were eluted with buffer N1 containing 300 mM imidazole. Ric8A1–492 and Ric8A1–452 were dialyzed against 20 mM phosphate buffer (pH 7.0) containing 5% glycerol, 5 mM β-mercaptoethanol (buffer S1). They were then further purified by SP-sepharose (GE healthcare) cation exchange chromatography. Resin was washed first with buffer S1 and then with buffer S1 containing 25 mM NaCl. Ric8A1–492 and Ric8A1–452 were eluted using buffer S1 containing 250 mM NaCl. Due to its poor solubility and/or stability at pH 7.0, Ric8A1–126 was purified using HiTrapQ anion exchange chromatography, as were Gα$_t$, Gα$_s$, and Gα$_i$. After purification using the His•Bind resin, these proteins were dialyzed against 20 mM HEPES (pH 8.0) buffer containing 100 mM NaCl, 5% glycerol, 1 mM tris(2-carboxyethyl)phosphine (TCEP) (buffer Q1) and loaded onto a HiTrapQ column (GE healthcare). Proteins were eluted using a linear 0.05–1 M NaCl gradient in buffer Q1. Where the His6-tag was not removed, Ric8A1–492, Ric8A1–452, Ric8A1–426, Gα$_t$, Gα$_s$, or Gα$_i$ was further purified by size-exclusion chromatography (SEC) using a HiLoad 16/600 Superdex 200 pg column equilibrated with 20 mM Tris, 150 mM KCl, 5% glycerol, 1 mM TCEP, pH 7.5. His tags were removed from Ric8A constructs by adding TEV protease at a 1:10 molar ratio, followed by an overnight incubation at 4 °C. Samples were then passed through His•Bind resin to remove uncleaved proteins and purified by SEC as described above.

The miniGα$_i$ construct was expressed and purified according to the procedures described above with the following modifications. Buffer N2 (50 mM HEPES, 150 mM NaCl, 20 mM MgSO4, 10% glycerol, pH 8.0) supplemented with 50 μM GDP was used to resuspend and sonicate the cells. His•Bind resin was charged with Co⁺⁺. After the column was loaded with the supernatant, it was washed with buffer N2 containing 10 mM imidazole and the protein was eluted with buffer N2 containing 150 mM imidazole and 50 μM GDP. The sample was then incubated overnight with TEV protease at a 1:50 molar ratio at 4 °C and purified by SEC using a HiLoad 16/600 Superdex 75 pg column equilibrated with 20 mM Tris-HCl (pH 8.0) buffer containing 150 mM KCl, 10% glycerol, 20 mM MgSO4, 10 μM GDP, and 1 mM TCEP.

Ric8A1–492/miniGα$_i$ complex was prepared by mixing Ric8A with miniGα$_i$ at a 1:1.5 molar ratio. The complex was purified by SEC using a HiLoad 16/600 Superdex 200 pg column equilibrated with 20 mM Tris-HCl (pH 8.0) buffer containing 150 mM KCl, 5% glycerol, and 1 mM TCEP. This procedure removed excess miniGα$_i$, ensuring 1:1 stoichiometry of the complex. Ric8A1–492/Gα$_t$ complexes were purified by mixing Ric8A1–492 with Gα$_t$ (1:1.5 molar ratio), followed by SEC using Superdex 200 10/300 GL column (GE healthcare) to remove excess Gα$_t$.

MBP-Gα$_t$327–350 was purified by Ni-NTA chromatography followed by SEC using a HiLoad 16/600 Superdex 200 pg column equilibrated with 20 mM Tris-HCl (pH 7.5) buffer containing 100 mM NaCl, 1 mM EDTA, and 2 mM TCEP. Ric8A1–492/MBP-Gα$_t$327–350 complexes were prepared by mixing the proteins at a 1:1.5 molar ratio in 5 mM maltose and loading the mixture onto a HiLoad 16/600 Superdex 200 pg column equilibrated with 20 mM Tris-HCl (pH 7.5) buffer containing 150 mM KCl, 5% glycerol, and 1 mM TCEP.

**Crystallization, data collection, and structure determination**. MBP-Gα$_t$327–350 was crystalized using the hanging drop vapor-diffusion method. Specifically, 0.3 μl of protein (60 mg/ml, 5 mM maltose) was mixed with 0.3 μl of crystallization solution containing 2 M ammonium sulfate, 0.1 M sodium acetate (pH 4.6) at 18 °C, using a TTP LabTech Mosquito crystallization robot (TTP labtech). Apo Ric8A1–492 was also crystallized using the hanging drop vapor-diffusion method. In this case 0.3 μl of Ric8A1–492 at 35 mg/ml was mixed with 0.3 μl of 0.1 M bis-tris propane (pH 6–7), 0.2 M NaI, 12–20% PEG3350 at 4 °C using a TTP LabTech Mosquito crystallization robot. Crystals obtained were used to seed 1:1 μl drop in a sitting drop vapor-diffusion set-up. Crystals were cryo protected using 20% sucrose in mother liquor. The Ric8A1–492/MBP-Gα$_t$327–350 complex was crystallized by mixing 0.3 μl at 40 mg/ml with 0.3 μl of 0.1 M MIB buffer, 25% PEG3350, pH 8.0 at 18 °C, using a TTP LabTech Mosquito crystallization robot.

The data sets were collected at the Advanced Light Source Beamline 4.2.2 (Berkley, CA). MBP-Gα$_t$327–350 crystals were exposed to the beam for 0.2 s with a wavelength of 1 Å at 0.2° oscillation per frame, and data were collected across a 180° rotation. For apo Ric8A1–492, crystals were exposed to the beam for 1 second, with 0.2° of oscillation per frame, and two data sets were collected across a 180° rotation. For crystals of the Ric8A1–492/MBP-Gα$_t$327–350 complex, two data sets were collected by exposing crystals to the beam for 0.1 s, with 0.1° of oscillation across a 180° rotation. The data sets were indexed and integrated using X-ray detector software XDS[54] and scaled using the Scala software[55]. For apo Ric8A and the Ric8A1–492/MBP-Gα$_t$327–350 complex, the two data sets were merged and the structures were solved by molecular replacement. To solve the structure of MBP-Gα$_t$327–350, PDB ID 1ANF was used as search model and molecular replacement was done using Phaser crystallographic software[56]. The structure of the Ric8A1–492/MBP-Gα$_t$327–350 complex was solved by molecular replacement using the MBP-Gα$_t$327–350 structure as a search model. The structure of apo

Ric8A1–492 was solved using the Ric8A structure from the complex as a search model. The structures were refined using PHENIX[57] and Coot. For MBP-Gα$_t$327–350, Ramachandran favored, allowed and outliers (%) were 99.23, 0.77, and 0. For Ric8A1–492/MBP-Gα$_t$327–350, Ramachandran favored, allowed and outliers (%) were 97.46, 2.42, and 0.13. For apoRic8A, Ramachandran favored, allowed and outliers (%) were 97.44, 2.05, and 0.51. Figures were generated using Pymol. Electrostatic surfaces were calculated using the APBS software[58].

**Protein thermostability assays**. Stabilities of Ric8A constructs with and without peptide ligands were assessed using differential scanning fluorimetry (DSF), in which an increase in fluorescence of the Sypro Orange dye is measured. Ric8A1–492, Ric8A1–452 and Ric8A1–426 in 20 mM Tris-HCl (pH 7.5) buffer containing 150 mM KCl, 5% glycerol, and 1 mM TCEP with and without peptide were used at a final concentration of 0.5 mg/ml and were supplemented with a 1000-fold dilution of Sypro Orange dye (Invitrogen). Temperature was increased at 1 °C/minute and fluorescence signals were recorded using real-time PCR (C1000 Touch thermal cycler, Bio Rad), with the cycler set to FRET mode and the excitation wavelength set at 450–490 nm and the emission wavelength at 560–580 nm.

**Bio-layer interferometry binding assay**. An Octet RED96 system and streptavidin (SA)-coated biosensors (FortéBio, Menlo Park, CA) were used to measure association and dissociation kinetics for Gα peptides or the full-length Avi-tagged Gα$_t$ in relation to Ric8A1–492, Ric8A1–452, and Ric8A1–426. To obtain the Avi-tagged Gα$_t$, Gα$_t$ was cloned into modified pET21a vector with the N-terminal His6-tag followed by the Avi tag and TEV cleavage site[59]. The Avi-tagged I340Q/N343H mutant of Gα$_t$ was prepared by QuikChange protocol. BL21(DE3) cells for expression of the Avi-tagged Gα$_t$ were grown in 2TY media supplemented with biotin (10 mg/liter), induced at OD$_{600}$ of 0.6 with 50 μM IPTG and further grown overnight at 16 °C. Binding studies were performed in 20 mM Tris, 150 mM KCl, 5% glycerol, 1 mM TCEP, 0.5 mg/ml BSA, pH 8.0. All steps were performed at 26 °C, with biosensors stirred into 0.2 ml of sample in each well at 1000 rpm, and at a data acquisition rate of 5.0 Hz. N-terminally biotinylated Gα$_t$333–350, Gα$_s$363–380, C1 peptide and C2 peptide were loaded onto SA sensors at a concentration of 0.05, 0.05, 0.005, and 0.01 mg/ml for 40–90 seconds. Data for association and dissociation phases of the assay were collected as shown in Fig. 1b and Supplementary Figs. 8, 11, 12, and 14. To correct for baseline drift and non-specific binding, reference sensors lacking bound Gα peptide were used in the BLI assays with Ric8A proteins at the highest concentrations. Kinetic data fitting was performed using FortéBio Data Analysis software 10.0. For each concentration of Ric8A1–492, dissociation rate constant ($k_d$) values were calculated from the corresponding dissociation phases of the curves. These $k_d$ values were used to calculate the association rate constant ($k_a$) values from the association phases for each concentration according to the equation

$$k_a = (k_{observed} - k_d)/[Ric8A1 - 492] \tag{1}$$

The average $k_a$ and $k_d$ were calculated as means of the individual $k_a$ and $k_d$ values for all curves. Equilibrium dissociation constant $K_D$ was calculated as mean $k_d$/ mean $k_a$. Steady-state data fitting was performed using the GraphPad Prism 7 software with the equation for one site specific binding.

**Crosslinking**. Apo Ric8A1–492, and complexes of Ric8A1–492 with Gα$_t$ and miniGα$_i$, were purified by SEC on a column equilibrated with 20 mM HEPES (pH 8.0) buffer containing 150 mM KCl, 5% glycerol and 1 mM TCEP. Crosslinking reactions were initiated by adding disuccinimidyl suberate (DSS) (0.5 mM final concentration) to apo Ric8A1–492 (0.3 mg/ml), Ric8A1–492/Gα$_t$ (0.3 mg/ml) or Ric8A1–492/miniGα$_i$ (0.35 mg/ml) at 25 °C. Forty minutes after being initiated, they were quenched by adding Tris-HCl pH 7.5 to a final concentration of 30 mM. Crosslinked proteins were resolved by SDS-PAGE.

**PAGE and in-gel trypsin digestion**. An estimated 3 μg of crosslinked protein was loaded onto NuPage 4–12% Bis-Tris precast gels (Invitrogen, USA) and separated at 150 V for 1.5 h. Sharp Pre-stained Protein Standards (10 μl, were loaded onto a separate gel lane to serve as a guide to molecular weight. The gel was stained using a Pierce Silver Stain Kit (Thermo Scientific, USA) following the manufacturer's directions.

A procedure slightly modified from the one described previously was used for in-gel digestion[60]. In brief, the targeted protein bands from the SDS-PAGE gel were manually excised, cut into 1 mm³ pieces, and washed in 100 mM ammonium bicarbonate:acetonitrile (1:1, v/v) and then in 25 mM ammonium bicarbonate: acetonitrile (1:1, v/v) to achieve complete destaining.

The gel pieces were further treated with acetonitrile, to effectively dry them and then reduced in 50 μl of 10 mM DTT at 56 °C for 60 min. The gel-trapped proteins were then alkylated with 55 mM chloroacetamide (CAM) for 30 min at room temperature. The gel pieces were washed twice with 25 mM ammonium bicarbonate:acetonitrile (1:1, v/v) to remove excess DTT and CAM, after which 50 μl of cold trypsin solution at 10 ng/μl in 25 mM ammonium bicarbonate was added to the gel pieces and they were allowed to swell on ice for 60 min. Digestion was conducted at 37 °C for 16 h. Peptides were extracted by adding 100 μl of 50% acetonitrile/0.1% formic acid for 0.5 h three times and combining the supernatants.

The combined extracts were concentrated using a lyophilizer and rehydrated in 15 μl of Mobile Phase A solution.

**LC-MS/MS.** Mass spectrometry data were collected using an Orbitrap Fusion Lumos mass spectrometer or an Q-Exactive HF Orbitrap mass spectrometer (Thermo Fisher Scientific, San Jose, CA) coupled to an Easy-nLC-1200™ System (Proxeon P/N LC1400). The autosampler was set to aspirate 3 μl (estimated 0.2 ug) of reconstituted digest and load the solution on a 2.5-cm C18 trap (New Objective, P/N IT100–25H002) coupled to waste, HV or analytical column through a microcross assembly (IDEX, P/N UH-752). Peptides were desalted on the trap using 16 μl mobile phase A in 4 min. The waste valve was then blocked and a gradient was run at a 0.4 μl/min flow rate through a self-packed analytical column (10 cm in length×75 μm inner diameter). The fused silica column was tapered from 75 μm ID (Polymicro) to ~8 μm at the tip using a Sutter P-2000 laser puller, and then packed with 2.7 μm Halo C18 particles using a He-pressurized SS cylinder. Peptides were separated in-line with the mass spectrometer using a 70-min gradient composed of linear and static segments wherein buffer A is 0.1% formic acid and buffer B is 95% acetonitrile, 0.1% formic acid. The gradient first holds at 4% for 3 min then makes the following transitions (%B, min): (2, 0), (35, 46), (60, 56), (98, 62), (98, 70).

**Tandem mass spectrometry using Orbitrap Fusion Lumos.** Data acquisition was initiated with a survey scan ($m/z$ 380–1800) acquired on an Orbitrap Fusion Lumos mass spectrometer at a resolution of 120,000 in the off axis Orbitrap segment (MS1), with Automatic Gain Control (AGC) set to 3E06 and a maximum injection time of 50 ms. MS1 scans were acquired every 3 s during the 70-min gradient described above. The most abundant precursors were selected from among 2–6 charge state ions at a 1E05 Automatic Gain Control (AGC) and 70 ms maximum injection time. Ions were isolated with a 1.6-Th window using the multi-segment quadrupole and subjected to dynamic exclusion for 30 sec if they were targeted twice during the prior 30-s period. The selected ions were then sequentially subjected to collision-induced dissociation (CID) and high energy collision-induced dissociation (HCD) activation in the IT and the ion routing multipole respectively (IRM). The AGC target for CID was 4.0E04, 35% collision energy, with an activation Q of 0.25 and a 75 ms maximum fill time. Targeted precursors were also fragmented by high energy collision-induced dissociation (HCD) at 30% collision energy in the IRM. HCD fragment ions were analyzed using the Orbitrap (AGC 1.2E05, maximum injection time 110 ms, and resolution set to 30,000 at 400 Th). Both MS2 channels were recorded as centroid and the MS1 survey scans were recorded in profile mode.

**Tandem mass spectrometry using Orbitrap Q-Exactive HF.** Data dependent acquisitions (DDA) began with a survey scan (m/z 380–1800) acquired on a Q-Exactive HF Orbitrap mass spectrometer at a resolution of 120,000 in the off axis Orbitrap segment (MS1) with AGC set to 3E06 and a maximum injection time of 50 ms. MS1 scans were acquired every 3 s during the 70 min gradient described above. The most abundant precursors were selected among 2–5 charge state ions observed in MS1 and isolated with a 1.6-Th window using the segment quadrupole. Selected ions were subjected to HCD using 1E05 AGC and 70 ms injection time thresholds. Ions were subject to dynamic exclusion for 30 s if they were targeted twice in the ion routing multipole (IRM) during the prior 30 sec. Targeted precursors were fragmented by HCD at 30% collision energy and fragment ions were analyzed using the Orbitrap (AGC 1.2E05, maximum injection time 110 ms, and resolution set to 30,000 at 400 Th). MS1 survey scans were recorded in profile mode and MS2 data were recorded as centroid.

**Identification of crosslinked peptides.** Peak lists in the form of mgf files were submitted for the search using Protein Prospector Batch-Tag Web[34,61]. The database searched contained the sequences of Ric8A1–492, Gα$_t$ or miniGα$_i$. In addition, the target sequences were randomized 10 times and appended to the target sequences. Eighty peaks from each spectrum were searched using a precursor charge range of 2–5, a tolerance of 20 ppm for precursor ions and 1 Da for fragment ions, and an instrument setting of ESI-Q-hi-res. Cleavage specificity was set to that of trypsin, and up to three missed cleavages per peptide were allowed. Carbamidomethylation of cysteines was specified as a constant modification. Protein Prospector Search Compare program was used to generate a Crosslinked Peptides report. The score of a crosslinked peptide was based on number and types of fragment ions identified, as well as the sequence and charge state of the cross-linked peptide. Only results where the score difference is >0 (i.e., the crosslinked peptide match was better than a single peptide match alone) are considered[61]. The expectation value represents how many random matches would be expected to achieve a given score or greater, in a search of a given size. The expectation values are calculated based on matches to single peptides and thus should be treated as another score, rather than a statistical measure of reliability[61]. There were no matches to decoy sequences in the search of intramolecular crosslinks of Ric8A1–492, suggesting a false discovery rate (FDR) of <1%. FDRs of ~4.5% and 1.5% were estimated in the Ric8A1–492/miniGα$_i$ and Ric8A1–492/Gα$_t$ crosslinking searches, respectively, as described previously[61]. Low-scoring intermolecular crosslink matches were filtered according to an FDR of 5%.

**GTPγS binding to Gα$_i$.** The rates of GTPγS binding to Gα$_i$ in the absence or presence of Ric8A1–492 (Ric8A1–452) were measured by following the increase in fluorescence of Gα$_i$ tryptophan. Gα$_i$ (1 μM) was mixed with GDP (1 μM) in a fluorescence cuvette, in 20 mM Tris-HCl (pH 8.0) buffer containing 150 mM NaCl, 5% glycerol, 1 mM TCEP and 10 mM MgCl$_2$, and incubated for 2 min. The binding reaction was initiated by addition of Ric8A1–492 or Ric8A1–452 (1 μM each) and/or and GTPγS (10 μM). Fluorescence at 340 nm was monitored with the excitation set at 295 nm. Data were fit to an equation for one phase association using the GraphPad Prism 7.05 software.

**Small angle X-ray scattering.** SAXS data were collected at the Bio-CAT beamline 18-ID-D at the Advanced Photon Source (APS; Argonne, IL) using an in-line size-exclusion chromatography SAXS (SEC-SAXS) configuration[62] with superdex 200 column (GE Healthcare). A 250-μl volume of 10 mg/ml sample in 20 mM Tris, 150 mM KCl, 5% glycerol, 1 mM TCEP, pH 7.5 buffer was loaded onto the column at flow rate at 0.9 ml/min. The elution trajectory was redirected into the SAXS sample flow cell (1.5 mm ID quartz capillary with 10 μm walls) after the UV monitor. Scattering data were collected every 2 s using a 0.5-s exposure on a Pilatus 3 × 1 M pixel detector (DECTRIS) covering a $q$-range of $0.0040 < q < 0.388$ Å$^{-1}$ ($q = 4\pi/\lambda \sin \theta$, where $2\theta$ is the scattering angle). For each protein, the buffer scattering before and after the eluted peak was recorded and used for background correction. The final protein scattering curves were obtained by scaling the data from the main peak, and averaging it and correcting for buffer scattering. BioXTAS RAW and ATSAS 2.8 were used for SAXS data reduction and analysis[63,64].

**Molecular dynamics simulations.** MD simulations were performed using YASARA Structure 18.2.7 and the md_runfast macro. For simulation of Ric8A1–426, the Ric8A1–492 structure solved in the P2$_1$ space group (PDB 6N85) was used as starting model. Missing residues from the loop regions were modeled using the YASARA Structure 18.2.7 before starting the simulation. For the simulation of Ric8A1–452, the top selected FloppyTail model was used. For simulation of miniGα$_i$, a homology model of miniGα$_i$ lacking the 25 N-terminal residues (ΔN25-miniGα$_i$) was built based on the template structure of miniGα$_s$ in complex with the β2 adrenergic receptor (PDB ID 5G53) using the YASARA program. This choice of template produced fewer clashes in modeling the complex of miniGα$_i$ with Ric8A compared to the use of miniGα$_o$ structure (PDB: 6FUF). The simulations were run using the AMBER14 force field in water at a temperature of 298 K or 310 K, pH of 7.4 and NaCl concentration of 0.9%. The particle mesh Ewald summation was used to compute long-range coulombic interactions with a periodic cell boundary and a cutoff of 8 Å. The MD simulations were analyzed using the md_analyze macro in YASARA and Pymol programs. All of the parameters of these MD simulations except for RMSF were calculated using YASARA. RMSF (Cα) values were calculated using VMD. Backbone atoms of residues corresponding to 80–280 of Ric8A was aligned before the RMSF calculations were performed using built in RMSF calculation function in VMD[65].

**Steered molecular dynamics simulations.** SMD was performed on a conformation of the Ric8A1–452 model that was derived in MD simulations and showed minimal clashes on modeling of the Ric8A1–452/ ΔN25-miniGα$_i$ complex. The structure file was prepared using VMD[65] and the plugin QwikMD[66]. The MD simulations were performed employing the NAMD molecular dynamics package[67] and the CHARMM36 force field[68]. The Minimization and Constrained equilibration MD Simulation was performed with implicit solvent represented by the Generalized Born/solvent-accessible surface area model[69,70]. A temperature ramp was performed and consisted of 0.24 ns of simulation where the temperature was raised from 60 K to 300.00 K. Before the SMD simulations all the systems were submitted to an energy minimization protocol for 1000 steps. In this step consisted of 1.00 ns of simulation, the atoms defined by the selection "protein and backbone" were restrained. The SMD simulation was performed with implicit solvent represented by the Generalized Born/solvent-accessible surface area model[69,70]. The temperature was maintained at 300.00 K using Langevin dynamics. A distance cutoff of 16.0 Å was applied to short-range, non-bonded interactions, and 15.0 Å for the smothering functions. The equations of motion were integrated using the r-RESPA multiple time step scheme[67] to update the short-range interactions every 1 steps and long-range electrostatics interactions every 2 steps. The time step of integration was chosen to be 2 fs for all simulations. The SMD simulations[71] of constant velocity stretching (SMD-CV protocol) employing a pulling speed of 2.5 Å/ns and a harmonic constraint force of 7.0 kcal/mol/Å$^2$ was performed for 4.0 ns. In this step, SMD was employed by harmonically restraining the position of Ric8A residues 1–13 and moving a second restraint residues 296–452 with constant velocity in the axis defined by the center of mass of the ΔN25-miniGα$_i$ atoms that clash with Ric8A and the center of mass of the Ric8A atoms that clash with ΔN25-miniGα$_i$. Residues 296–452 were selected as moving because in MD simulations of Ric8A1–452 they behaved as a module that fluctuated relatively independent of the rest of the molecule. It is this module that was clashing with ΔN25-miniGα$_i$.

**Modeling of the proximal C terminal tail of Ric8A.** The crystal structure of Ric8a1–426 was used as the starting model in modeling of the proximal portion of the C-terminal tail. Given that the regular secondary structure of the core domain

ends at residue 422, the starting model was generated based on the structure of Ric8A1–422 with Ric8A423–452 attached in a random conformation that extended away from the core of the molecule. The structures of the missing loops were modeled using the YASARA program, and the resulting structure was used in further modeling with the FloppyTail application[35] of the Rosetta software suite. The FloppyTail algorithm generates hypothetical, low-energy conformations for disordered or flexible regions using two-stage modeling. The first stage is the centroid phase, during which side-chains are represented by a single, large centroid atom and is designed to collapse long tails into a reasonable conformation. The second stage involves restoration of the side-chains, fine sampling of the backbone conformational space, side-chain optimization, and minimization[35,72]. The various options for the FloppyTail algorithm were set using a flag file (provided as Supplementary Note 1). The flag file was modified from the original flag file provided by Steven Lewis as part of the Rosetta software suite. Two experimental distance constraints based on the highest-scoring C-terminal crosslinks K408/K449 and K352/K449 identified by XL-MS were used during the FloppyTail calculation with flat harmonic function (eq. 2)[73].

$$f(\text{dist}) = \begin{cases} 0 & \text{if dist} \leq \text{tolerance} \pm x_0 \\ \left(\frac{\text{dist} - x_0 - \text{tolerance}}{\sigma}\right)^2 & \text{otherwise} \end{cases} \quad (2)$$

The three parameters chosen for the calculation were $x_0 = 15$ Å, tolerance = 15 Å and $\sigma = 1$. Effectively, these parameters limit the distance between C$\alpha$ atoms to <30 Å. The length of the DSS crosslinker (11.4 Å) combined with the length of 2 Lys side-chains (6.4 + 6.4 Å) yields a maximal distance of ~24 Å between the C$\alpha$–C$\alpha$ atoms. Considering protein dynamics (flexibility) this distance is adjusted to a threshold of <30 Å[74]. Simulations were performed using 56 cores on the Argon cluster at the University of Iowa. After the FloppyTail simulation, Rosetta score_jd2 executable was used to calculate energy scores of the models, and the Crysol program[75] was used to generate and compare fits of theoretical SAXS profiles of the models to experimental SAXS data ($\chi^2$ values). Energy scores from Rosetta and $\chi^2$ were used to select the models. The range of $\chi^2$ values among the 500 top energy models was 1.13–5.00. As a first step in model selection, 311 models were picked from the top 500 energy score models using the cutoff $\chi^2 < 2.0$. Next, the pool of models was further narrowed to 212 using the crosslinking distance constraint of <30 Å[74] for the C$\alpha$ atoms of the third highest scoring C-terminal crosslinked pair K375/K449; this constraint was not used in the FloppyTail modeling. Clustering of 212 models was performed with the Ensemble Cluster tool from UCSF Chimera software[36], and it yielded two main clusters I and II that were comprised of 50 and 20 models, respectively. The remaining clusters were minor each containing 13 or fewer models. None of the minor clusters included models from the top 10 energy score models. Thus, these clusters totaling 142 models were excluded from further analysis.

**Modeling of apo Ric8A1–492**. The starting model for apo Ric8A1–492 was built using the YASARA application and the model of Ric8A1–452, which served as a template, and was subjected to a conformational sampling analysis using the BILBOMD server[37]. Residues Ric8A1–452 and an α-helix Ric8A 471–490 (Supplementary Fig. 6) were treated as rigid domains linked with a flexible linker Ric8A453–470. Thus, 800 BILBOMD models were generated using MD simulations and validated against the experimental SEC-SAXS profile of Ric8A1–492 (Supplementary Fig. 19).

**Modeling of Ric8A-miniG$\alpha_i$ and Ric8A-G$\alpha_t$ complexes**. The starting model for the complex was produced by superimposition of the α5 helix of the MD models of ΔN25-miniG$\alpha_i$ onto the corresponding helical segment of the C terminal G$_t$ peptide in complex with Ric8A, which was aligned with the SMD models of Ric8A1–452. The SMD model of Ric8A1–452 and MD model of ΔN25-miniG$\alpha_i$ that produced no clashes on this superimposition were selected, and their coordinates were merged. Next, the N-terminus of ΔN25-miniG$\alpha_i$ in the complex was extended by 6 residues (ΔN19-miniG$\alpha_i$) using YASARA to include the crosslinked Lys21 residue of miniG$\alpha_i$. Also, the C-terminal CGLF residues of ΔN19-miniG$\alpha_i$ were modeled as in the Ric8A-bound structure of G$\alpha_t$327–350. The resulting model of the complex was energy minimized using YASARA and used as input for the FloppyTail calculation. The FloppyTail algorithm was used to model the distal C terminus of Ric8A in the Ric8A1–492/ΔN19-miniG$\alpha_i$ complex. The model of the Ric8A1–452/ΔN19-miniG$\alpha_i$ complex with the Ric8A453–492 tail appended in the extended conformation was used as a starting point for this modeling. Residues 471–490 were kept helical, based on the prediction by PSIPRED (Supplementary Fig. 6). The FloppyTail protocol for the complex was similar to that described for Ric8A1–452. Two experimental intermolecular distance constraints were used during the simulation. 5548 models generated by FloppyTail were clustered into two major clusters I and II by RMSD with a cutoff of 2.5 Å using the clustering algorithm implemented in the visual molecular dynamics (VMD) program. The top cluster I and II models (models 1 and 2, respectively) in terms of energy score were selected for the 19-residue N-terminal extension to generate the models of Ric8A1–492/miniG$\alpha_i$ complex.

The structure of rhodopsin-bound G$_i$ (PDB 6CMO) was used to model of the Ric8A/G$\alpha_i$ complex. Additionally, the αN-helix and the α5 helix of G$\alpha_i$ from 6CMO were modeled according to the model of the Ric8A1–492/miniG$\alpha_i$ complex. The model of the Ric8A1–492/G$\alpha_t$ was obtained by superimposing the G$\alpha_i$

structure onto the model of the Ric8A1–492/miniG$\alpha_i$ complex followed by energy minimization using the YASARA program.

**Reporting summary**. Further information on research design is available in the Nature Research Reporting Summary linked to this article.

## Data availability
The atomic coordinates have been deposited in the Protein Data Bank (https://www.rcsb.org/) with the PDB accession codes 6N84[10.2210/pdb6n84/pdb], 6N85[10.2210/pdb6N85/pdb], 6N86[10.2210/pdb6n86/pdb]. SAXS data for Ric8A1–492 and Ric8A1–452 were deposited in the Small Angle Scattering Biological Data Bank (https://www.sasbdb.org/) with the accession codes SASDF65[https://www.sasbdb.org/data/SASDF65/] and SASDF75[https://www.sasbdb.org/data/SASDF75/], respectively. The coordinates for the models of Ric8A1–452, apo Ric8A1–492, Ric8A1–492/miniGα complex, and Ric8A1–492/Gα complex are provided as Supplementary Data 5, 6, 7, and 8. The source data underlying Figs. 1a–c, 3f–h, 5a, f and Supplementary Figs. 8, 9, 10, 12, 13, 14B, 15 and 23 are provided as a Source Data file. All other data supporting the findings of this study are available from the corresponding author upon reasonable request.

## Code availability
Rosetta Floppy tail scripts used in this study are available in Supplementary Note 1.

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

## Acknowledgements

This work was supported by the National Institutes of Health grant RO1 EY-12682 to N.O. A. We thank Christine Blaumueller for discussions, Jay Nix (Molecular Biology Consortium 4.2.2 beamline at Advanced Light Source) for aid in remote data collection,

Srinivas Chakravarty (BioCAT facility, Advanced Photon Source) for help in SAXS data collection, and Nicholas Schnicker (Protein and Crystallography Facility at the University of Iowa) for assistance in BLI data collection. This research used resources of the Advanced Light Source, a Department of Energy Office of Science user facility under Contract DEAC02–05CH11231, and resources of the Advanced Photon Source, a U.S. Department of Energy (DOE) Office of Science User Facility operated for the DOE Office of Science by Argonne National Laboratory under Contract No. DE-AC02–06CH11357 and supported by grant 9 P41 GM103622 from the National Institute of General Medical Sciences of the National Institutes of Health. Use of the Pilatus 3 1M detector was provided by grant 1S10OD018090–01 from NIGMS. We would like to acknowledge use of resources at the Carver College of Medicine's Protein and Crystallography Facility at the University of Iowa. Mass spectrometry data were collected in the University of Iowa Proteomic facility directed by Dr. R.M. Pope, supported by an endowment from the Carver Foundation and by a Thermo Lumos awarded by an HHMI grant to Dr. Kevin Campbell.

## Author contributions

Conceptualization by D.S. and N.O.A.; methodology by D.S., L.G. and N.O.A.; investigation by D.S., L.G. and N.O.A.; writing by D.S. and N.O.A.; funding acquisition by N.O.A.

## Additional information

**Competing interests:** The authors declare no competing interests.

**Peer Review Information**: *Nature Communications* thanks the anonymous reviewers for their contribution to the peer review of this work. Peer reviewer reports are available.

