## [Peer Review File · Nature Communications]

Reviewers' comments:

Reviewer #1 (Remarks to the Author):

This manuscript presents a novel work that describes the structural underpinnings of the Ric8 protein and its interactions with the G α subunit of heterotrimeric G-proteins. First, the crystal structures of the apo and holo configurations of a truncated Ric8A in complex with G α variants were solved. Then, biochemical analyses in the form of thermal stability assays were carried out to determine interactions between the two species. Cross-linking mass spectrometry was then performed, providing important structural details regarding the proximity of regions not able to be captured by crystallography. The authors then performed a series of modelling and molecular dynamics simulations to realize a model of the unknown Ric8A C-terminal domain.

While this constitutes a substantial structural and functional characterization body of work, of a no doubt important class of proteins, revisions will be required to streamline data presentation and the conclusions made. The reviewer recognizes the difficulty of the work. However it feels like the manuscript is preliminary and substantial revision may be required. Moreover, the authors may consider revisiting their manuscript to ensure that the text is clear, and figures are of high quality and free of errors. This may include both the main text, figures, figure legends and any supplementary documents.

Major comments:

- The authors have performed a series of molecular dynamics simulations enabling model structures of the Ric8A complex – the C-terminal domain of which lacks ordered structure and has evaded attempts of solution via crystallography. The reviewer would like to see included, additional descriptions of the simulations performed, their simulation durations, and supplementary figures including a breakdown of both energetic and dimensional parameters of their simulated systems. This should include energy, temperature, pressure, RMSD, RMSF, radius of gyration and solvent accessible surface area (SASA) over time, for all simulations performed. The authors should take additional care to ensure that all units are included in descriptions of the simulations and in figures and legends.
- The reviewers would appreciate greater transparency on the modelling performed in this manuscript, regarding the 4900 initial models generated of the Ric8A C-terminal domain, and the 800 BILBOMD models from MD simulations. Since 4900 models were generated and only two such models are shown in Figure 4, the reviewers would like to see additional analyses regarding the

variation across the model ensemble. This could include pairwise RMSD data, presented in the form of an RMSD matrix, displaying the variation of models generated from this study, and/or clustering analysis performed to identify subpopulations of particular model conformations. The reviewers would also appreciate greater transparency with regards to model selection, and suggest that the authors clearly state how models were ranked and based on what criteria.

- Regarding the cross-linking mass spectrometry carried out in this study, the reviewers commend the use of cross-links as modelling restraints, which can compensate for unknowns from other techniques. The reviewer however, feel that the authors should take extra care in their preparation of cross-link data and figures. Figures 4E and 5A-C do not reach their potentials in demonstrating the merits of this study. The authors are suggested to appropriately label both their proteins and cross-links in all figures, ensure that the text is legible and that the main text is complimented with appropriate descriptions of the figures. The authors should also include a clarified breakdown of how the cross-links were used in modelling, include the length of the physical cross-linker molecule, what chemical groups/sidechains it targets, and how distance measurements were performed and what the distances were for all modelling.

- In the cross-linking section, and Figure 4E, authors have collected eleven cross-links in the C-terminal region of the Ric8A protein, and seven of these are visibly projected onto the structure of the protein in Figure 4E. Where are the other cross-links? It also appears strange that cross-links were not captured in any other areas of the protein. Authors should provide additional information regarding the quality of their cross-linking results, what distances were measured (including what atoms measurements were taken from) and justify their use of a 30Å distance threshold.

- Authors suggest that “Thus, the crosslinking analysis is consistent with the notion that electrostatic interactions between the negatively charged proximal C-terminal segment Ric8A427-452 and the positively charged surface on the Ric8A core domain play an important role in stabilizing Ric8A.” Have the authors considered why if the C-terminal region of Ric8A is electrostatically interacting with the core domain, why such a seemingly stabilizing interaction is not reciprocated in their crystal structures? Are these non-specific electrostatic interactions?

- Authors should justify their use of certain proteins within this study, for example, the results section introduces Gat327-350, Gat333-350 and Gat340-350, yet there is no explanation of why these particular lengths were used. Authors should also avoid making unjustified jumps in the main text such as where the authors have made a sudden change from Gat327-350 to Gat333-350:

“The structure of the Ric8A1-492/MBP-Gat327-350 complex indicates that the flexible C-terminal region Ric8A427-492 is not directly involved in binding the Gα C-terminus. However, this region is known to contain a potential Gα binding site. To determine the role of Ric8A427-492 in the function

of this protein, we first used BLI to assess an interaction between Ric8A1-426 (lacks the C-terminal region) and Gat333-350.”

- The authors first present biochemical data in Figure 4A to show that the Gat333-350 peptide has a stabilizing effect on the Ric8A1-492 protein. 4900 loop models were then generated and evaluated on their quality – two of these were selected based on them having the ‘best scores’. The two models are presented in Figure 4C-D, and readers are reassured that these show good agreement between theoretical and experimental scattering curves. However, given that the two models are so different, their scattering profiles appear nearly identical in Figure 4B. How different are the scattering profiles and structures of the other 4900 models generated? The authors should provide some indication of the RMSD spread across the population of models, perhaps even perform clustering to identify major subpopulations of structures and provide additional supplementary figures to show this information appropriately.

Minor comments:

- Many undefined acronyms within the abstract: ‘Ric8A’, ‘GPCR’, ‘GEF’, ‘Gat’ and ‘ARM’
- Authors should consistently use ‘G-protein’ or ‘G protein’
- Authors should maintain consistent use of figure labels, e.g. ‘Sup. Fig.’, ‘Suppl. Fig.’ and ‘Supplementary Table’, are used.
- Manuscript should be checked for typographical errors, e.g. ‘Ric8aA’
- Please could authors ensure that all acronyms are explained in figure legends – there is no explanation of ‘RFU’ for Figure 1.
- Authors should check that font sizes are appropriately used, e.g. Figure 2A and 3D.
- The legend of Figure 2 is not formatted correctly.
- Include error bars for all measurements performed in triplicate.
- Authors should avoid using a superscript ‘0’ and underlined ‘+’ in place of ° and ±.
- ARM repeats of Ric8A are named as R1-R8, yet in Figure 2A, where the author is directed to, the repeats are again inconsistently labelled as ARM1-8.
- (h) is not required after “three α -helices”.
- Remove the word ‘entire’ in “Although almost the entire Gat sequence (residues 327-348) could be traced in electron density” – this is redundant as the residues are listed after. Using entire in this context ambiguously suggests the whole intact Gat sequence.
- Please could authors clarify the sentence: ‘The binding and dissociation kinetics for the Ric8A1-492/Gat333-350 interaction were consistent with a 1:1 binding model, with the global

association constant (k_{on}) measured at $1.34 \times 10^5 \text{ M}^{-1}\text{s}^{-1}$ and the average dissociation constant (k_d) at 0.031 s^{-1} , yielding a K_D of $0.23 \text{ }\mu\text{M}$ (Fig. 1C).'

- Figure 1D is entirely missing a figure legend.
- The authors should include at least some representation or view of all three crystal structures followed by general descriptions. In the current version, Figure 2A shows the structure of Ric8A extracted from the Ric8A-Gat327-350 complex while the authors possess a structure of the similar intact apo Ric8A.
- Authors claim that "However, the limited structural similarity (root mean square deviation, RMSD, $> 3 \text{ }\text{\AA}$) does not extend beyond ~ 400 N-terminal residues of Ric8A." It is unclear where this 3 A calculation comes from.
- It is unclear how the surface area calculation of "... revealed an interface that burries $956 \text{ }\text{\AA}^2$ of surface area" was performed.
- Suggest to the authors to change the colours of Ric8A and Gat peptide in Figure 3A and 3B – it is not immediately apparent that there are two proteins. Authors should also fully label the contents of their figures where possible, both to avoid ambiguity and shorten the lengths of the figure legends.
- Authors make statements such as "However, it is possible that the contacts made by D333 and A334 of Gat in the crystal were influenced by the need to accommodate MBP in the lattice" without providing any figure to show these interactions.
- For the statement "The C terminal carboxylate group of Gat F350 forms a hydrogen bond and polar interactions with the side-chain amide of N123, as well as the guanidino groups of R71 and R75 of Ric8A (Fig. 3D)", authors need to show the electron density for these residues at an appropriate contour level and quote this in the figure legend. It is possible that these sidechains are in these positions only because of crystallographic refinement.
- R71 and R75 referred to in-text, are found as R70 and R74 in Figure 3D.
- "For instance, Gat I339 and I340 occupy a hydrophobic patch formed by Ric8A F169, F228 and F232." This needs to be shown in a figure.
- Supplementary Figure 7 requires reworking. Panel B is referred to before panel A in the main text. It is unclear what the lengths of the schematic represent and how the schematic was generated.
- In the following line "This analysis revealed that the truncated protein .." is ambiguous since Gat327-350 has been truncated to Gat333-350 and Ric8A1-492 has been truncated to Ric8A1-426 under the same section.
- Authors do not explain the graphs shown in Suppl. Fig. 10A or how they reach the conclusion that "Unexpectedly, the Gat C-terminal peptide did not bind to Ric8A1-426".

- “Both the thermal stability of Ric8A1-452 and its ability to bind to and be stabilized by Gat333-350 were comparable to those of Ric8A1-492, indicating that proximal C-terminal residues 427-452 are essential (Fig. 4A, Suppl. Fig. 11).” This sentence focuses on deciphering the importance of residues 427-452 in determining whether 333-350 can be bound by Ric8A. The data used to reach this conclusion is presented as supplementary data, while the supporting comparison of 1-492:333-350 is presented as a main figure.
- The method of calculating the SAXS scattering profiles is missing from the manuscript.
- Authors state that “Residues K349, K352, and K408 are part of a large and highly conserved positively charged patch on the outer surface of the C-terminal portion of the Ric8A core domain (Fig. 4C,D)” yet none of these residue are labelled in the figures referred to.
- Authors claim that “Furthermore, model 1 is more consistent than model 2 with the cross-linking constraints for the crosslinked pair K375/K449” but provide no evidence to show how this conclusion was reached.
- “Based on our findings from XL-MS analysis, two harmonic distance constraints for the crosslinked pairs K408/K449 and K352/K449 were applied (Suppl. Fig. 12).” Authors describe that five cross-links were found in the C-terminal region of the protein. Could authors clarify why not all of these were used for their modelling, but only the two between K408/K449 and K352/449.
- “was that the C-terminal residues flanked the Ric8A356371 loop from opposite sides (Fig. 4C,D; Suppl. Fig. 12).” It is not clear what the authors mean by this sentence, the figure that the authors refer to does not aid interpretation.
- “The top apo Ric8A1-492 model based on SAXS suggests that the distal portion of the Cterminal tail is in an extended conformation and does not contact the core of the protein (Fig. 4E, Suppl. Fig. 13).” How was this observation made? One way to support this would be to compare the experimental Dmax of the protein obtained from SAXS P(r) curves with length measurements made from simulations

Reviewer #2 (Remarks to the Author):

Srivastava et al. describes structure/function studies of Ric8A and its interactions with peptides derived from the C-termini of heterotrimeric G proteins. Two key pieces of data provided are crystal structures of unliganded Ric8A and of Ric8A in complex with MBP fused to the G protein C-terminal peptide. Luckily, MBP stabilizes the peptide interaction by binding to an acidic region near the C-terminus of Ric8A. Unluckily this seems to displace the C-tail of Ric8A which is known to be important for GEF activity and selectivity and this region is consequently not observed in the structure (although dynamic to begin with). The unliganded structure is low resolution, but this is

okay because it is chiefly used to demonstrate a lack of large conformational change upon complex formation with the C-terminal peptide. The complex structure is high resolution and the model seems of high quality. Modeling an intact G protein alpha subunit based on the peptide suggested to the authors that there must be a conformational change in either the G protein or Ric8A to avoid steric overlap, which they model via steered molecular dynamics. The fact that receptor bound structures of Galpha subunits (nucleotide free) dock better than nucleotide bound forms suggests that Ric8A is preferentially binding to a nucleotide free conformation of the G protein, as would be expected. The authors show via binding experiments with different Galpha C-termini that the interactions with the C-terminus of Galpha contain some, but not all possible selectivity determinants (Gi vs. Gs). Finally, to map how the C-terminus of Ric8A might be driving GEF activity, they conduct cross linking studies followed by mass spectrometry and show that basic residues in this element within bind to an acidic region (masked by MBP in the structure) to stabilize the fold, and that there are additional crosslinks by other residues in the C-terminus with elements near switch II of the alpha subunit. Such interactions with the nucleotide binding pocket of the G protein might explain why the C-terminus of Ric8A is required for GEF activity (as opposed to binding). It may also explain other avenues for selectivity.

Key Strengths:

- This could be the first reported structure of Ric8A, either with or without bound ligand. So, first in class structure. First in class complex.
- Gives detailed, high resolution description of how the C-termini of Galpha subunits interact with the concave surface of Ric8A, which likely represents the major binding determinant for this complex.
- Some new insights, although lower resolution, into how the extreme C-terminus of Ric8A might be involved in GEF activity on the bound G protein subunit.

Key Weaknesses:

- We learn primarily about one key binding determinant of Ric8A and not as much about the molecular basis for its activity.
- No experimental validation of the role of key residues in crystallographically observed protein interface or interfaces presumed via cross linking studies. This seems particularly important when one only has a fragment of the entire protein substrate bound.
- MD models are highly speculative when not experimentally validated
- Poor execution of the writeup

Major editorial comments:

The key experimental weakness to this manuscript is the lack of experimental validation of the G protein complex, modeled based on the crystallographic data. It is unfortunate that the important C-terminus of Ric8A is not observed in the crystallographic work, but this may have been required to get any data at all. Because only a fragment of Galpha is bound, it seems pretty important to at least mutate residues on Ric8A that they observe to interact with the peptide to test that indeed they are important for binding intact Galpha subunits. Crosslinking data is also highly artifactual, especially with respect to flexible elements, and thus it is essential that the residues involved in the crosslinks, or those in the proposed contact regions presumed from the cross linking data (because the cross linked residues are not necessarily the interacting ones in the native complex) be tested experimentally.

The coordinates for hypothetical docked models should also be provided for consumption by the general readership in the supplemental materials.

The quality of the paper is also significantly affected by the manner in which it is written. For example, sometimes extensive boring method-like text is found in the results sections. In some places the language comes across more like a sales pitch than an objective study. A lot of this is detailed below in the below section. Each instance on its own is a minor flaw, but this work would have been a lot more compelling if the manuscript were more tidy and better annotated in the figures.

Minor editorial comments (in order of occurrence):

Intro. The authors should avoid use of terminology/language like "in exquisite detail" and "tantalizing parallels" because such descriptors are scientifically meaningless and are typically used more to impress the reader than accurately describe the system. What should "tantalizing" mean to an experimentalist anyhow (used again in the discussion)? What if I am not quite as excited as you are? Just stick to the facts and let them do the work here and I will decide for myself if it is tantalizing or not.

Page 4. " and stabilization of the conformation in which RD and HD are separated, facilitating the escape of GDP." Not technically accurate. The consensus in the field is that there is not just one

conformation of the G protein when RD and HD are separated. The two domains become more independent and can have multiple conformations.

Fig. 1B. This DLS data adds nothing to the paper over the DSF in panel A and I would recommend omitting, especially because one cannot derive a T_m from it (no saturation).

Fig. 1C. The values or units need attention. "25 50 100 250 100 250 nM"?

Page 6/7: "Our analysis indicates that the interaction between Ric8A and the $G\alpha$ C-terminus is of markedly higher affinity than previously estimated." It is okay to say this, but please state what the prior value was so we can easily evaluate how "markedly" it is.

BLI affinity measurements: These were derived from fitting a one phase binding curve to the increase in response (nm) vs. Ric8A concentration. However, I wonder if they would get a less markedly different value from previously reported if they evaluation the K_{on} and K_{off} data intrinsic to the data shown in panel 1C. This is because the "response" may not truly reflect an equilibrium driven value. Usually these experiments are usually taken to saturation as well, which was not achieved at least in Fig. 1.

Page 7. Cut to the chase and move all the phasing structure determination details to the methods. It is not important that one used MR to solve the structures, or the space groups, or even the R-factors. There was nothing unusual in the approach either. All that information is in tables and the methods. Instead immediately dig into the key pieces of structural data that the crystal structures reveal. Of course tell us what they are first, but all this out of place methodology is a real mood killer.

Page 8. "with the exception of a single stretch of residues (471-489) that exhibit a strong propensity to form an α -helix (Suppl. Fig. 6)." I would be careful about saying things like "strong propensity". Many things that show up in secondary structure prediction programs to be helical turn out to be otherwise. Unless you experimentally test the program, I would make no assumptions. After all, even the C-terminus of $G\alpha$, which we know likes to be helical, is disordered in solution. Thus one might say its real propensity is to not have secondary structure at all. It is a matter of context. I guess my main objection here is that you are letting a program say what the answer is, but it needs to be tested first.

Page 9. "Ric8A. By binding across the concave surface of Ric8A, the G α C-terminus could act as a scaffolding element that bridges multiple ARMs." The fact that the structure shows the peptide binding across multiple ARMs (as stated above in text) seems to make this obvious. Omit?

Page 16. Typo: "In models 3 and ,..". The next paragraph "To Model..." needs to all be moved to the methods. Another example of methods in the results.

Page 16. The results section seems to end abruptly here. So a model was generated. What did we learn? Or was this methods section meant to cover the prior paragraphs in the results?

Discussion. Delete "Agonist-bound GPCRs trigger activation of heterotrimeric G proteins that transduce

signals and control critical processes including transcription, cell division, and cell migration 39-41.

Ric8A is a major player in several of these pathways as well. Unlike the GPCRs, it acts only on the G α subunits, when they are not complexed with G $\beta\gamma$, and it functions as both a GEF and a chaperone 6,7,42. Despite the crucial cellular roles of Ric8A, its structure and molecular mechanisms have been elusive." Redundant with introduction.

Page 19. "Unexpectedly, our studies have revealed profound and unique roles". As you might guess by now, I object to vague descriptors like "profound" and "unique" or even "unexpectedly". It seems well established that the C-terminus of Ric8A is important for function, so why is it unexpected that you found some interesting roles for it? Plus you describe it as unstructured, yet in the models you install a helix based on secondary structure prediction algorithms? Seems contradictory.

Crystal Structures/data: These look good. The suggestion I have is for you to build out all the residues, even the ones for which you don't see all the density for, as opposed to truncating them to alanine. An example is residue M386 in the complex structure, which is a lysine but modeled as an alanine. The Fo-Fc maps suggest that there is more side chain present than just a C β , and if one contours lower one can see pretty convincing density the length of a lysine. It is better to let the B-factors blur out conformationally heterogeneous atoms than to effectively set their occupancy to 0 (via truncation). Plus it better reflects reality.

Required: In the main text provide a Fo-Fc omit map of the bound C-terminal G α peptide. This is a key piece of data that shows your maps are not phase biased. I am sure it will look great.

Fig 1 caption (and elsewhere). Seems unlikely that you know K_{on} to 3 significant figures. Check all the reported data values in paper for proper accuracy.

Fig 2 and 3 (and elsewhere): would be helpful to have cartoon "turn arrows" to graphically show how the molecules are related to each other rather than describing in the caption.

Modeling: Nucleotide free G_s (from β_2 receptor complex) was used for a lot of the modeling. However, nucleotide free G_t (and G_i) are now available. These would have been more appropriate for these studies.

Fig 2. Please label N and C termini in all the panels, and use a different font size from the panel labels.

Fig 3. This and other figures seem under annotated. N and C termini of both Ric8A and bound peptide, for example. What do the colors correspond to in panel c, for example? What is the scale?

Fig 4. Panel E is not possible to interpret.

SAXS data. There is minimal SAXS data provided. Although authors are not using to generate envelopes, basic data reporting is necessary, such as raw scattering curve, Guinier plot, and $P(r)$ plot. Concentration dependence too if using static SAXS. See Gao Westfield et al. 2017 JBC.

Fig 5. These are hard to interpret and under annotated. Maybe use different colors rather than arrows? What am I supposed to get out of panel C?

Supplemental:

Page 4: Typo: "for xx, xx, xx, and xx seconds"

Fig S1: What does "tight" mean and can one really say that based on SEC alone anyhow? Label bands in gels with the proteins they are believed to be. Give MW estimates in the top chromatograms for the peaks. Again, annotation is weak in figures overall.

Fig. S2. Minimal caption and annotation.

Fig S3. What is the extra band on the far right side of the gel? Looks to be another lane.

Fig S4. Be better if annotated and Ric8A were held in the same orientation in each comparison.

Fig S5. Turn arrows would be useful. However, it is hard to imagine how progression from panel a to c is just a 90° turn of Ric8A.

Fig S6. Not high enough quality and not sure it is important enough to include anyhow. Omit?

Fig S7. Misleading because the squiggly lines could be interpreted as disordered or no secondary structure, and in some of these structure the extreme C-terminus is truly disordered, others not but has no regular secondary structure. The fact that the Galpha C-terminus is conformationally heterogeneous is not a strong point of the paper anyhow, so I think this whole figure could be omitted. Not surprising it does something weird in the MBP fusion structure.

Fig S8. This figure seems like it should be in the main text. Although labeled/annotated.

Fig S14. As above, strange that nucleotide-free Gi or Gt wasn't used for this exercise.

Fig S15. Replace "essential" with "important"? Looks like one gets an uptick in exchange in 1-452, so it might not be "essential"....

Fig S16. Would be helpful to label the bands.

Table 1. Adjust significant figures throughout.

Table 2. This is a data dump, and it would be more accessible to the readers if it were curated to distill out the most important data. As it is it seems to be the log file from the server run.

Reviewer #3 (Remarks to the Author):

Srivastava et al. solved the crystal structures of the resistance to inhibitors of cholinesterase 8A (Ric8A) protein: apo and in complex with the C-terminal fragment of G α t attached to a maltose-binding protein (MBP) tag. In addition to X-ray crystallography the authors used a number of complimentary methods which are described in detail in the manuscript. Perhaps the level of detail is too high - being a non-expert it was hard to read.

My expertise lies in small angle scattering, not within guanine nucleotide exchange factors, therefore I will comment only on the use of SAXS in the manuscript. The de facto standard "2017 publication guidelines for structural modelling of small-angle scattering data from biomolecules in solution" by Trewhella et al. *Acta D* 73(9):710–728 recommend that "SAS data and models be deposited and made freely available in a public data bank [currently there is SASBDB and BIOISIS]. Ideally q , $I(q)$ with standard errors should be deposited for each measured profile and the associated models plus details of how the experiment was conducted with the data and model validation parameters and analyses". Depositing the SAXS data presented in Fig. 4B (Ric8A1-452) and Suppl. Fig. 13 (Ric8A1-492) would allow to assess their quality and the accuracy of the FloppyTail models. Further Trewhella et al. recommend to describe the conduct and results of SEC–SAXS experiments in a table consisting of the following sections: (a) sample and buffer details; (b) SAXS data collection parameters; (c) software employed for SAXS data reduction, analysis and interpretation; (d) structural parameters; (e) modelling. Addition of such a table to the supplementary materials would strengthen the presentation of the SAXS analysis.

On page 13 and in the Fig. 4B caption the authors report "Chi" values. A "chi-squared" value appears to be more appropriate to describe how well a computed scattering pattern fits the observed experimental SAXS data.

It should be mentioned what program was used to calculate the theoretical SAXS profiles from the models presented in Fig. 4C.

On page 7 it is mentioned that only Ric8A residues 1-422 "could be modeled based on the electron density, suggesting that the C-terminal tail of Ric8A is flexible". A typical approach to interpreting SAXS data from partially flexible proteins would be using an ensemble of models instead of a single model, however the fits in Fig. 4B and in Suppl. Fig. 13 appear to be from one model each - it is not clear why (the authors even cite the relevant paper Pelikan et al. 2009). Fitting each SAXS data set with an ensemble of models instead of a single model could improve the presented fits.

A minor issue: Fig. 4A caption says "Denaturation of 9.1 μ M Ric8A1-452" but in the figure we read "Ric8A1-492".

We have thoroughly amended our manuscript, including figures, according to the comments from the Reviewers, and have responded to each below. (Reviewers' comments are italicized.)

Reviewer #1 (Remarks to the Author):

Major comments:

- *The authors have performed a series of molecular dynamics simulations enabling model structures of the Ric8A complex – the C-terminal domain of which lacks ordered structure and has evaded attempts of solution via crystallography. The reviewer would like to see included, additional descriptions of the simulations performed, their simulation durations, and supplementary figures including a breakdown of both energetic and dimensional parameters of their simulated systems. This should include energy, temperature, pressure, RMSD, RMSF, radius of gyration and solvent accessible surface area (SASA) over time, for all simulations performed. The authors should take additional care to ensure that all units are included in descriptions of the simulations and in figures and legends.*

We have included all of the requested parameters for molecular dynamics (MD) simulations of Ric8A1-426 and the model of Ric8A1-452 in Suppl. Fig. 15. The parameters of the MD simulation of mini-Gai and steered MD (SMD) simulation of Ric8A1-452 are now reported in Suppl. Figures 18 and 19, respectively.

- *The reviewers would appreciate greater transparency on the modelling performed in this manuscript, regarding the 4900 initial models generated of the Ric8A C-terminal domain, and the 800 BILBOMD models from MD simulations. Since 4900 models were generated and only two such models are shown in Figure 4, the reviewers would like to see additional analyses regarding the variation across the model ensemble. This could include pairwise RMSD data, presented in the form of an RMSD matrix, displaying the variation of models generated from this study, and/or clustering analysis performed to identify subpopulations of particular model conformations. The reviewers would also appreciate greater transparency with regards to model selection, and suggest that the authors clearly state how models were ranked and based on what criteria.*

We now describe our modified algorithm for ranking Ric8A1-452 models and selecting of the top model (**lines 259-280; lines 734-736**). The final pool of 212 models was selected from 4900 models based on the energy scores from the FloppyTail modeling, filtering models using their agreement with the SAXS profile ($\chi^2 < 2.0$), and an additional crosslinking constraint K375/K449 < 30Å (not used in the modeling). This pool of models was clustered using the Ensemble Cluster tool from UCSF Chimera software. Representative top energy models from the two largest clusters have been added as a new Supplemental Fig. 14. The best energy ranking model from the largest cluster (which is also the overall top energy model in the pool of 212 models) was selected as the Ric8A1-452 model (previously referred to as model 1). We omitted description of model 2 from the manuscript for clarity and due to its relatively poor χ^2 value.

The output from the BILBOMD server does not include detailed parameters of MD simulations. Rather it outputs a model or model ensembles as pdb files and theoretical SAXS profiles of models including their fits (χ values) with experimental SAXS profiles. For general details of the BILBOMD modeling the readers are referred to the original paper (**lines 739-742**). We

added the top 3-state model (Suppl. Fig. 16D) to illustrate the range of conformational flexibility of the C-terminal tail in models of apo Ric8A.

- *Regarding the cross-linking mass spectrometry carried out in this study, the reviewers commend the use of cross-links as modelling restraints, which can compensate for unknowns from other techniques. The reviewer however, feel that the authors should take extra care in their preparation of cross-link data and figures. **Figures 4E and 5A-C** do not reach their potentials in demonstrating the merits of this study. The authors are suggested to appropriately label both their proteins and cross-links in all figures, ensure that the text is legible and that the main text is complimented with appropriate descriptions of the figures. The authors should also include a clarified breakdown of how the cross-links were used in modelling, include the length of the physical cross-linker molecule, what chemical groups/sidechains it targets, and how distance measurements were performed and what the distances were for all modelling.*
- *In the cross-linking section, and Figure 4E, authors have collected eleven cross-links in the C-terminal region of the Ric8A protein, and seven of these are visibly projected onto the structure of the protein in Figure 4E. Where are the other cross-links? It also appears strange that cross-links were not captured in any other areas of the protein. Authors should provide additional information regarding the quality of their cross-linking results, what distances were measured (including what atoms measurements were taken from) and justify their use of a 30Å distance threshold.*

We have revised the figures showing the crosslinking results. We have included a zoomed-in view of the areas of interest, labeled crosslinked residues and colored crosslinks used to constrain the modeling differently. We used disuccinimidyl suberate (DSS) as a crosslinker, which crosslinks lysine side-chain amino groups. The length of the crosslinker (11.4 Å) combined with the length of 2 Lys side-chains (6.4+6.4 Å) yields a maximal distance of ~ 24 Å between the Ca-Ca atoms. Considering protein dynamics (flexibility) this distance is adjusted to a threshold of <30 Å according to a study by Merkley et al. The crosslinking distance constraint of <30 Å was used for all modeling. The above justification for the distance constraint is now included in the Methods section (**lines 726-729**).

The majority of the Ric8A intramolecular crosslinks were indeed in the C-terminal part of the molecule. This is likely because the C-terminal half of Ric8A-1-492 contains many more Lys residues (14 Lys) compared to the N-terminal half (5 Lys). However, there was a crosslink K44/K48 found in the N-terminal part of the protein.

- *Authors suggest that “Thus, the crosslinking analysis is consistent with the notion that electrostatic interactions between the negatively charged proximal C-terminal segment Ric8A427-452 and the positively charged surface on the Ric8A core domain play an important role in stabilizing Ric8A.” Have the authors considered why if the C-terminal region of Ric8A is electrostatically interacting with the core domain, why such a seemingly stabilizing interaction is not reciprocated in their crystal structures? Are these non-specific electrostatic interactions?*

In the complex crystal lattice, the Ric8A core is stabilized by contacts between its positively charged region and the negatively charged domain of MBP (Suppl. Fig. 5). This packing interaction appears to displace the negative patch of the C-terminal tail, Ric8A436–444 (EDEDTDDE) from binding to the positively charged surface. Thus, the interaction is not reciprocated in this crystal structure. In the structure of apo Ric8A the C-terminal tail is also not resolved. It is likely, that the interaction of the C-terminal tail with the core is dynamic, and it is allosterically and reciprocally enhanced upon binding of the Gα C-terminus to the core.

Nonetheless, we believe that the interaction is specific. This notion is supported by the strong conservation of the residues involved. Furthermore, recent study suggests that Ser435 and Thr440 are phosphorylated and this augments Ric8A function (Papasergi-Scott, et al.). Considering our data, we suggest that phosphorylation of Ser435 and Thr440 enhances the stabilizing electrostatic interactions with the positive core patch and thereby potentiates Ric8A activities. We now discuss this potential mechanism for the effect of Ric8A phosphorylation on its function in the Discussion section (**lines 396-400**).

- *Authors should justify their use of certain proteins within this study, for example, the results section introduces Gat327-350, Gat333-350 and Gat340-350, yet there is no explanation of why these particular lengths were used. Authors should also avoid making unjustified jumps in the main text such as where the authors have made a sudden change from Gat327-350 to Gat333-350:*

“The structure of the Ric8A1-492/MBP-Gat327-350 complex indicates that the flexible C-terminal region Ric8A427-492 is not directly involved in binding the $G\alpha$ C-terminus. However, this region is known to contain a potential $G\alpha$ binding site. To determine the role of Ric8A427-492 in the function of this protein, we first used BLI to assess an interaction between Ric8A1-426 (lacks the C-terminal region) and Gat333-350.”

At the start of our studies, it was known that the Gai C-terminal 18-mer peptide binds to Ric8A. This peptide corresponds to Gat333-350, which we used for our studies. A longer sequence Gat327-350 was used in the MBP-fusion protein to allow for a linker, such that potential crystal packing interactions would not interfere with the known interaction involving Gat333-350. In retrospect, this choice of the MBP construct and the Gat333-350 peptide was quite opportune, since only residues Gat333-350 are seen interacting with Ric8A in the Ric8A1-492/MBP-Gat327-350 complex, and only the contacts made by D333 and A334 of Gat in the crystal were possibly influenced by the need to accommodate MBP into the lattice. The 11-mer Gat340-350 was used in the fusion protein and as peptide, since it has been shown to contain main selectivity determinants of the $G\alpha$ /rhodopsin interaction (PDB 6CMO) and its GRCP-bound structure has been solved.

We now include our rationale for the constructs in the Methods section, since the main text is limited to 5,000 words (**lines 444-452**).

- *The authors first present biochemical data in Figure 4A to show that the Gat333-350 peptide has a stabilizing effect on the Ric8A1-492 protein. 4900 loop models were then generated and evaluated on their quality – two of these were selected based on them having the ‘best scores’. The two models are presented in Figure 4C-D, and readers are reassured that these show good agreement between theoretical and experimental scattering curves. However, given that the two models are so different, their scattering profiles appear nearly identical in Figure 4B. How different are the scattering profiles and structures of the other 4900 models generated? The authors should provide some indication of the RMSD spread across the population of models, perhaps even perform clustering to identify major subpopulations of structures and provide additional supplementary figures to show this information appropriately.*

See our response above regarding Ric8A1-452 model selection and clustering. We now use χ^2 values rather than χ as this is a more commonly accepted criterion. The range of χ^2 values among the top 500 energy score models was 1.13-5.00. Given the low resolution of SAXS (≥ 20 Å), we used a cutoff value of $\chi^2 \leq 2.0$, which generally signifies good agreement of

a theoretical SAXS curve for a model with experimental SAXS data. Applying this cutoff value yielded 311 models out of 500 (**lines 268-270**).

Minor comments:

- Many undefined acronyms within the abstract: 'Ric8A', 'GPCR', 'GEF', 'Gat' and 'ARM'
- Authors should consistently use 'G-protein' or 'G protein'
- Authors should maintain consistent use of figure labels, e.g. 'Sup. Fig.', 'Suppl. Fig.' and 'Supplementary Table', are used.
- Manuscript should be checked for typographical errors, e.g. 'Ric8aA'
- Please could authors ensure that all acronyms are explained in figure legends – there is no explanation of 'RFU' for Figure 1.
- Authors should check that font sizes are appropriately used, e.g. Figure 2A and 3D.
- The legend of Figure 2 is not formatted correctly.

We have made all of the appropriate corrections to the text

- Include error bars for all measurements performed in triplicate.

We now show average curves instead of representative curves for the DSF stability experiments (Figures 1A, 6A, Suppl Figures 7 and 11; n=3 or 4). For the BLI experiments we do not normalize the data so that the actual collected data are better reflected in the manuscript. For these experiments we show representative data curves. However, relevant average values (T_m , K_D) are reported as Mean \pm SE (n=3) in the figures or figure legends.

- Authors should avoid using a superscript '0' and underlined '+' in place of ° and \pm .
- ARM repeats of Ric8A are named as R1-R8, yet in Figure 2A, where the author is directed to, the repeats are again inconsistently labelled as ARM1-8.
- (h) is not required after "three α -helices".
- Remove the word 'entire' in "Although almost the entire Gat sequence (residues 327-348) could be traced in electron density" – this is redundant as the residues are listed after. Using entire in this context ambiguously suggests the whole intact Gat sequence.

We have made all of the appropriate corrections to the text

- Please could authors clarify the sentence: 'The binding and dissociation kinetics for the Ric8A1-492/Gat333-350 interaction were consistent with a 1:1 binding model, with the global association constant (k_{on}) measured at $1.34 \times 10^5 M^{-1}s^{-1}$ and the average dissociation constant (k_d) at $0.031 s^{-1}$, yielding a KD of $0.23 \mu M$ (Fig. 1C).'

We replaced "global" with "average". Also, k_{on} is replaced with k_a for clarity. k_{on} is frequently used together with k_{off} , whereas analogous k_a is typically used with k_d . The meaning of the sentence should become clearer with the following description of our calculations of k_a , k_d and K_D , which is added to the Methods section. "For each concentration of Ric8A1-492, dissociation rate constant (k_d) values were calculated from the corresponding dissociation phases of the curves. These k_d values were used to calculate the association rate constant (k_a) values from the association phases for each concentration according to the equation $k_a = (k_{observed} - k_d) / [Ric8A1-492]$. The average k_a and k_d values were calculated as means of the individual k_a and k_d

values for all curves. In the kinetic analysis, the equilibrium dissociation constant K_D was calculated as mean k_d /mean k_a .” (lines 558-564)

- *Figure 1D is entirely missing a figure legend.*

This has been corrected.

- *The authors should include at least some representation or view of all three crystal structures followed by general descriptions. In the current version, Figure 2A shows the structure of Ric8A extracted from the Ric8A-Gat327-350 complex while the authors possess a structure of the similar intact apo Ric8A.*

The overviews of each structure in the study are shown in Suppl. Fig. 2 (complex), Fig. 2C (apo Ric8A) and Suppl. Fig. 9A (MBP-G α 327-350). All statistics for the structures are in Suppl. Table 1. We show the structure of Ric8A from the complex in Fig. 2A because the complex structure was solved first and at higher resolution than the apo Ric8A structure.

- *Authors claim that “However, the limited structural similarity (root mean square deviation, RMSD, > 3 Å) does not extend beyond ~400 N-terminal residues of Ric8A.”. It is unclear where this 3 Å calculation comes from.*

We indicated in the text that “Comparison of the Ric8A1-426 structure to those of other proteins in the Protein Data Bank” was performed “using DALI” server and cite an appropriate reference (lines 150-152). Outputs from this server include RMSD values.

- *It is unclear how the surface area calculation of “.. revealed an interface that burries 956 Å² of surface area” was performed.*

We now provide a reference indicating that the surface area was determined using the PDBePISA server (line 165).

- *Suggest to the authors to change the colours of Ric8A and Gat peptide in Figure 3A and 3B – it is not immediately apparent that there are two proteins. Authors should also fully label the contents of their figures where possible, both to avoid ambiguity and shorten the lengths of the figure legends.*

We have labeled Figs 3A and 3B as suggested. We retained the rainbow coloring of Ric8A in these figures as it helps to follow the orientation of the Ric8A molecule, particularly in Fig. 3B.

- *Authors make statements such as “However, it is possible that the contacts made by D333 and A334 of Gat in the crystal were influenced by the need to accommodate MBP in the lattice” without providing any figure to show these interactions.*

We now highlight these residues in Suppl. Fig. 2

- *For the statement “The C terminal carboxylate group of Gat F350 forms a hydrogen bond and polar interactions with the side-chain amide of N123, as well as the guanidino groups of R71 and R75 of Ric8A (Fig. 3D)”, authors need to show the electron density for these residues at an appropriate contour level and quote this in the figure legend. It is possible that these sidechains are in these positions only because of crystallographic refinement.*

We now show the omit maps for these residues in Fig. 3E.

- *R71 and R75 referred to in-text, are found as R70 and R74 in Figure 3D.*

This has been corrected (**lines 171-172**).

- *“For instance, Gat I339 and I340 occupy a hydrophobic patch formed by Ric8A F169, F228 and F232.” This needs to be shown in a figure.*

The residues involved in these contacts are now shown and labeled in Fig. 3E.

- *Supplementary Figure 7 requires reworking. Panel B is referred to before panel A in the main text. It is unclear what the lengths of the schematic represent and how the schematic was generated.*

The text has been revised to refer to Panel A first (Suppl. Figure 9 in the revised manuscript) (**lines 194-196**). Although the lengths of the schematic roughly reflect the length of the corresponding segments in Ga, it is not the main point of the schematic, which is to illustrate comparable extensions of the Ga C-terminal α -helix upon Ga binding to Ric8A and GPCRs

- *In the following line “This analysis revealed that the truncated protein ..” is ambiguous since Gat327-350 has been truncated to Gat333-350 and Ric8A1-492 has been truncated to Ric8A1-426 under the same section.*

We have updated the text so that “the truncated protein” is replaced with “Ric8A1-426” (**line 236**)

- *Authors do not explain the graphs shown in Suppl. Fig. 10A or how they reach the conclusion that “Unexpectedly, the Gat C-terminal peptide did not bind to Ric8A1-426”.*

For all concentrations of Ric8A1-426, the BLI response curves are at the level of instrument noise, indicating no binding between Ric8A1-426 and Gat333-350. This clarification is added to the Suppl. Figure 11 legend.

- *“Both the thermal stability of Ric8A1-452 and its ability to bind to and be stabilized by Gat333-350 were comparable to those of Ric8A1-492, indicating that proximal C-terminal residues 427-452 are essential (Fig. 4A, Suppl. Fig. 11).” This sentence focuses on deciphering the importance of residues 427-452 in determining whether 333-350 can be bound by Ric8A. The data used to reach this conclusion is presented as supplementary data, while the supporting comparison of 1-492:333-350 is presented as a main figure.*

We corrected the labeling error in the figure (revised Fig. 5A), which shows the data for Ric8A1-452

- *The method of calculating the SAXS scattering profiles is missing from the manuscript.*

We indicated in the Methods section that “the Crysol program was used to compare fits of theoretical SAXS profiles of the models to experimental SAXS data” (**lines 731-732**).

- *Authors state that “Residues K349, K352, and K408 are part of a large and highly conserved positively charged patch on the outer surface of the C-terminal portion of the Ric8A core domain (Fig. 4C,D)” yet none of these residue are labelled in the figures referred to.*

We now show a close-up view of this patch with labeled residues in Suppl. Fig. 5D

- *Authors claim that “Furthermore, model 1 is more consistent than model 2 with the cross-linking constraints for the crosslinked pair K375/K449” but provide no evidence to show how this conclusion was reached.*

Model 2 has been omitted from the manuscript (see comments on the model selection above).

- *“Based on our findings from XL-MS analysis, two harmonic distance constraints for the crosslinked pairs K408/K449 and K352/K449 were applied (Suppl. Fig. 12).” Authors describe that five cross-links were found in the C-terminal region of the protein. Could authors clarify why not all of these were used for their modelling, but only the two between K408/K449 and K352/449.*

The two crosslinks between K408/K449 and K352/449 are the highest confidence crosslinks as evidenced by the Scores and Score Difference values in the Protein prospector report. Two other crosslinks 349/449 and 408/446 are similar to the above crosslinks as K349 is in proximity to K352, and K446 is in proximity to K449. During the FloppyTail modeling, we noticed that overly constraining the modeling (i.e. using more than two distance constraints) results in some models having unrealistic conformations of the modelled portion. Thus, we chose to use constraints K408/K449 and K352/449 to generate our models, but the third-highest scoring C-terminal crosslink K375/K449 was used to select of models that satisfy the Ca-Ca <30Å. We added appropriate clarifications to the Methods and Results sections (**lines 270-272, 718-721, 733-735**).

- *“was that the C-terminal residues flanked the Ric8A356371 loop from opposite sides (Fig. 4C,D; Suppl. Fig. 12).” It is not clear what the authors mean by this sentence, the figure that the authors refer to does not aid interpretation.*

This sentence has been deleted.

- *“The top apo Ric8A1-492 model based on SAXS suggests that the distal portion of the Cterminal tail is in an extended conformation and does not contact the core of the protein (Fig. 4E, Suppl. Fig. 13).” How was this observation made? One way to support this would be to compare the experimental Dmax of the protein obtained from SAXS P(r) curves with length measurements made from simulations*

We specified in the text that the distal portion refers to residues Ric8A453-492 (**line 287**). The P(r) curve for apo Ric8A1-492 is now presented in Suppl. Fig. 16B, and the Dmax is stated in the legend. An extended conformation of this distal C-terminal tail in apo Ric8A1-492 is evident from the top 1-state and 3-state BILBOMD models shown in the new Suppl. Fig. 16.

Reviewer #2 (Remarks to the Author):

Key Strengths:

- *This could be the first reported structure of Ric8A, either with or without bound ligand. So, first in class structure. First in class complex.*
- *Gives detailed, high resolution description of how the C-termini of Galpha subunits interact with the concave surface of Ric8A, which likely represents the major binding determinant for this complex.*
- *Some new insights, although lower resolution, into how the extreme C-terminus of Ric8A might be involved in GEF activity on the bound G protein subunit.*

Key Weaknesses:

- *We learn primarily about one key binding determinant of Ric8A and not as much about the molecular basis for its activity.*
- *No experimental validation of the role of key residues in crystallographically observed protein interface or interfaces presumed via cross linking studies. This seems particularly important when one only has a fragment of the entire protein substrate bound.*
- *MD models are highly speculative when not experimentally validated*
- *Poor execution of the writeup*

We generally concur with the weaknesses stated above. To address the key validation concern, we tested four mutants of Ric8A designed to disrupt the interface of Ric8A with the Ga C-terminus that was observed in the crystal structure (**lines 183-191**). This new analysis is presented in Fig. 3F and Suppl. Figures 7 and 8, and it validates the interface. We also agree that the interfaces presumed from crosslinking and FloppyTail /MD models are speculative. These models lack the resolution necessary to conduct targeted validation by mutational analysis. However, considering the roles of the Ric8A C-terminal tail for its function and the lack of atomic information on its conformation, we believe that our analysis and modeling of the Ric8A C-terminal tail and the complex of Ric8A with Ga are essential. Furthermore, our models are consistent with existing biochemical evidence, and with the newly uncovered role for the proximal portion of the Ric8A C-terminal tail in protein stability. We have edited the manuscript according to the reviewer's suggestions.

Major editorial comments:

The key experimental weakness to this manuscript is the lack of experimental validation of the G protein complex, modeled based on the crystallographic data. It is unfortunate that the important C-terminus of Ric8A is not observed in the crystallographic work, but this may have been required to get any data at all. Because only a fragment of Galpha is bound, it seems pretty important to at least mutate residues on Ric8A that they observe to interact with the peptide to test that indeed they are important for binding intact Galpha subunits. Crosslinking data is also highly artifactual, especially with respect to flexible elements, and thus it is essential that the residues involved in the crosslinks, or those in the proposed contact regions presumed from the cross linking data (because the cross linked residues are not necessarily the interacting ones in the native complex) be tested experimentally.

See our response above

The coordinates for hypothetical docked models should also be provided for consumption by the general readership in the supplemental materials.

We included coordinate files for the models of Ric8A1-453, apo Ric8A1-492, Ric8A1-492/miniGa complex, and Ric8A1-492/Ga complex as Supplemental Material.

The quality of the paper is also significantly affected by the manner in which it is written. For example, sometimes extensive boring method-like text is found in the results sections. In some places the language comes across more like a sales pitch than an objective study. A lot of this is detailed below in the below section. Each instance on its own is a minor flaw, but this work would have been a lot more compelling if the manuscript were more tidy and better annotated in the figures.

We edited the manuscript and revised the figures for improved presentation and annotation.

Minor editorial comments (in order of occurrence):

Intro. The authors should avoid use of terminology/language like "in exquisite detail" and "tantalizing parallels" because such descriptors are scientifically meaningless and are typically used more to impress the reader than accurately describe the system. What should "tantalizing" mean to an experimentalist anyhow (used again in the discussion)? What if I am not quite as excited as you are? Just stick to the facts and let them do the work here and I will decide for myself if it is tantalizing or not.

We have edited the text as requested (**lines 60-62**)

Page 4. " and stabilization of the conformation in which RD and HD are separated, facilitating the escape of GDP." Not technically accurate. The consensus in the field is that there is not just one conformation of the G protein when RD and HD are separated. The two domains become more independent and can have multiple conformations.

This has been corrected as requested. (**lines 69-72**).

Fig. 1B. This DLS data adds nothing to the paper over the DSF in panel A and I would recommend omitting, especially because one cannot derive a T_m from it (no saturation).

The DLS data have been omitted.

*Fig. 1C. The values or units need attention. "25 50 100 250 **100** 250 nM"?*

This typo is corrected to "25 50 100 250 500 750 nM"

Page 6/7: "Our analysis indicates that the interaction between Ric8A and the G α C-terminus is of markedly higher affinity than previously estimated." It is okay to say this, but please state what the prior value was so we can easily evaluate how "markedly" it is.

The previously reported value (K_D 12 μ M) is now included (**lines 123-125**)

BLI affinity measurements: These were derived from fitting a one phase binding curve to the increase in response (nm) vs. Ric8A concentration. However, I wonder if they would get a less markedly different

value from previously reported if they evaluate the K_{on} and K_{off} data intrinsic to the data shown in panel 1C. This is because the "response" may not truly reflect an equilibrium driven value. Usually these experiments are usually taken to saturation as well, which was not achieved at least in Fig. 1.

The association curves are approaching saturation sufficiently to estimate the equilibrium values from the curve fitting. In our analysis, the K_D value from the kinetics k_d/k_a (or k_{off}/k_{on}) = 0.24 μ M (Fig. 1B) matches well with the K_D from the steady state calculation (K_D 0.27 μ M) (Fig. 1C). Furthermore, the steady-state data fit very well with hyperbola (goodness of fit $R^2=0.99$)

Page 7. Cut to the chase and move all the phasing structure determination details to the methods. It is not important that one used MR to solve the structures, or the space groups, or even the R-factors. There was nothing unusual in the approach either. All that information is in tables and the methods. Instead immediately dig into the key pieces of structural data that the crystal structures reveal. Of course tell us what they are first, but all this out of place methodology is a real mood killer.

The details of structure determination are moved to the Methods section or omitted as requested (**lines 532-535**, Suppl. Table 1).

Page 8. "with the exception of a single stretch of residues (471-489) that exhibit a strong propensity to form an α -helix (Suppl. Fig. 6)." I would be careful about saying things like "strong propensity". Many things that show up in secondary structure prediction programs to be helical turn out to be otherwise. Unless you experimentally test the program, I would make no assumptions. After all, even the C-terminus of Galpha, which we know likes to be helical, is disordered in solution. Thus one might say its real propensity is to not have secondary structure at all. It is a matter of context. I guess my main objection here is that you are letting a program say what the answer is, but it needs to be tested first.

The word "strong" is deleted and the sentence is re-worded to emphasize that this is just a prediction, not a known propensity. (**lines 160-161**)

Page 9. "Ric8A. By binding across the concave surface of Ric8A, the Gat C-terminus could act as a scaffolding element that bridges multiple ARMs." The fact that the structure shows the peptide binding across multiple ARMs (as stated above in text) seems to make this obvious. Omit?

We believe that this point, although obvious to expert readers, is worth emphasizing.

Page 16. Typo: "In models 3 and ,..". The next paragraph "To Model..." needs to all be moved to the methods. Another example of methods in the results.

Page 16. The results section seems to end abruptly here. So a model was generated. What did we learn? Or was this methods section meant to cover the prior paragraphs in the results?

We deleted most of the last paragraph in the Results section as it is redundant with the information in the Methods section and replaced it with new text. The model of the Ric8A1-492/G α complex in Fig. 6C is revised to include the structure of Gai based on PDB 6CMO. At the end of the results section, we focus on the range of positions available to the HD domain in the model (**lines 343-349**).

Discussion. Delete "Agonist-bound GPCRs trigger activation of heterotrimeric G proteins that transduce signals and control critical processes including transcription, cell division, and cell migration 39-41.

Ric8A is a major player in several of these pathways as well. Unlike the GPCRs, it acts only on the Galpha subunits, when they are not complexed with Gβγ, and it functions as both a GEF and a chaperone 6,7,42. Despite the crucial cellular roles of Ric8A, its structure and molecular mechanisms have been elusive." Redundant with introduction.

This text has been deleted as requested.

Page 19. "Unexpectedly, our studies have revealed profound and unique roles". As you might guess by now, I object to vague descriptors like "profound" and "unique" or even "unexpectedly". It seems well established that the C-terminus of Ric8A is important for function, so why is it unexpected that you found some interesting roles for it? Plus you describe it as unstructured, yet in the models you install a helix based on secondary structure prediction algorithms? Seems contradictory.

This has been edited as requested . **(lines 386-387)**.

Crystal Structures/data: These look good. The suggestion I have is for you to build out all the residues, even the ones for which you don't see all the density for, as opposed to truncating them to alanine. An example is residue M386 in the complex structure, which is a lysine but modeled as an alanine. The Fo-Fc maps suggest that there is more side chain present than just a Cbeta, and if one contours lower one can see pretty convincing density the length of a lysine. It is better to let the B-factors blur out conformationally heterogeneous atoms than to effectively set their occupancy to 0 (via truncation). Plus it better reflects reality.

We have now modeled all the side chains of all of the residues with visible backbones in the electron density map.

Required: In the main text provide a Fo-Fc omit map of the bound C-terminal G alpha peptide. This is a key piece of data that shows your maps are not phase biased. I am sure it will look great.

The omit map is now shown in Fig. 3D.

Fig 1 caption (and elsewhere). Seems unlikely that you know K_{on} to 3 significant figures. Check all the reported data values in paper for proper accuracy.

It is indeed incorrect to use 3 significant figures for k_{on} (or k_a) when K_D is shown with 2 significant figures . We corrected the numbers **(lines 120-121)** and checked other reported data.

Fig 2 and 3 (and elsewhere): would be helpful to have cartoon "turn arrows" to graphically how the molecules are related to each other rather than describing in the caption.

The figures are revised as requested

Modeling: Nucleotide free Gs (from b2 receptor complex) was used for a lot of the modeling. However, nucleotide free Gt (and Gi) are now available. These would have been more appropriate for these studies.

Three structures of GPCR-bound miniG α were available, miniG α s bound A2A receptors (5G53) and miniG α o bound to the HT1B receptor or bound to rhodopsin (PDB: 6FUF). Our models of miniG α i based on miniG α s and miniG α o were very similar. However, we chose to use miniG α s, because the slight difference in the tilting of its C-terminus compared to miniG α o reduces clashes with Ric8A at the start of modeling. We now indicate this in the Methods section (**lines 672-676**). Thus, avoiding additional clashes with the use of the G α s template required less MD/SMD manipulations. Tilting of the G α C-terminus likely depends on the bound receptor/partner. The tilting that reduces clashes is expected to be induced upon G α binding to Ric8A. For the model of the complex of the full-length G α with Ric8A, we now use the G α i structure from its complex with rhodopsin (PDB 6CMO) with the C-terminus of G α _i modeled according to the model of the Ric8A/miniG α _i complex.

Fig 2. Please label N and C termini in all the panels, and use a different font size from the panel labels.

Fig 3. This and other figures seem under annotated. N and C termini of both Ric8A and bound peptide, for example. What do the colors correspond to in panel c, for example? What is the scale?

Fig 4. Panel E is not possible to interpret.

The figures are revised as requested

SAXS data. There is minimal SAXS data provided. Although authors are not using to generate envelopes, basic data reporting is necessary, such as raw scattering curve, Guinier plot, and P(r) plot. Concentration dependence too if using static SAXS. See Gao Westfield et al. 2017 JBC.

We added the Guinier plots in Fig. 5B and Suppl. Fig. 16A, and the P(r) plot in Suppl. Fig. 16B

Fig 5. These are hard to interpret and under annotated. Maybe use different colors rather than arrows? What am I supposed to get out of panel C?

The figure is revised as requested

Supplemental:

Page 4: Typo: "for xx, xx, xx, and xx seconds"

This has been corrected. (**line 554**)

Fig S1: What does "tight" mean and can one really say that based on SEC alone anyhow? Label bands in gels with the proteins they are believed to be. Give MW estimates in the top chromatograms for the peaks. Again, annotation is weak in figures overall.

We have deleted the word "tight."

Fig. S2. Minimal caption and annotation.

Fig S3. What is the extra band on the far right side of the gel? Looks to be another lane.

Fig S4. Be better if annotated and Ric8A were held in the same orientation in each comparison.

Fig S5. Turn arrows would be useful. However, it is hard to imagine how progression from panel a to C is just a 90° turn of Ric8A.

The figures have been revised as requested.

Fig S6. Not high enough quality and not sure it is important enough to include anyhow. Omit?

We replaced the image.

Fig S7. Misleading because the squiggly lines could be interpreted as disordered or no secondary structure, and in some of these structure the extreme C-terminus is truly disordered, others not but has no regular secondary structure. The fact that the Galpha C-terminus is conformationally heterogeneous is not a strong point of the paper anyhow, so I think this whole figure could be omitted. Not surprising it does something weird in the MBP fusion structure

We believe that this figure would be helpful to readers. the main point of the schematic is to illustrate comparable extensions of the Ga C-terminal α -helix upon Ga binding to Ric8A and GPCRs. We added clarification to the figure legend (revised Suppl. Fig. 9).

Fig S8. This figure seems like it should be in the main text. Although labeled/annotated.

The figure is moved to the main text

Fig S14. As above, strange that nucleotide-free Gi or Gt wasn't used for this exercise.

See our response above.

Fig S15. Replace "essential" with "important"? Looks like one gets an uptick in exchange in 1-452, so it might not be "essential"....

This has been corrected.

Fig S16. Would be helpful to label the bands.

We have labeled the bands as requested.

Table 1. Adjust significant figures throughout.

This has been corrected.

Table 2. This is a data dump, and it would be more accessible to the readers if it were curated to distill out the most important data. As it is it seems to be the log file from the server run.

We think that this table is a useful source for readers to obtain instant information on the sequence conservation in Ric8 proteins, not only for the Ga C-terminus contact residues, but also for the essential acidic patch and the positively charged surface interacting with the patch. We deleted the similar Suppl. Table 3 that was generated for the Ric8A subtree.

Reviewer #3 (Remarks to the Author):

Srivastava et al. solved the crystal structures of the resistance to inhibitors of cholinesterase 8A (Ric8A) protein: apo and in complex with the C-terminal fragment of Gat attached to a maltose-binding protein (MBP) tag. In addition to X-ray crystallography the authors used a number of complimentary methods which are described in detail in the manuscript. Perhaps the level of detail is too high - being a non-expert it was hard to read.

My expertise lies in small angle scattering, not within guanine nucleotide exchange factors, therefore I will comment only on the use of SAXS in the manuscript. The de facto standard "2017 publication guidelines for structural modelling of small-angle scattering data from biomolecules in solution" by Trewthella et al. Acta D 73(9):710–728 recommend that "SAS data and models be deposited and made freely available in a public data bank [currently there is SASBDB and BIOISIS]. Ideally q , $I(q)$ with standard errors should be deposited for each measured profile and the associated models plus details of how the experiment was conducted with the data and model validation parameters and analyses". Depositing the SAXS data presented in Fig. 4B (Ric8A1-452) and Suppl. Fig. 13 (Ric8A1-492) would allow to assess their quality and the accuracy of the FloppyTail models. Further Trewthella et al. recommend to describe the conduct and results of SEC–SAXS experiments in a table consisting of the following sections: (a) sample and buffer details; (b) SAXS data collection parameters; (c) software employed for SAXS data reduction, analysis and interpretation; (d) structural parameters; (e) modelling. Addition of such a table to the supplementary materials would strengthen the presentation of the SAXS analysis.

We have deposited the SAXS data to Small Angle Scattering Biological Data Bank and added a table to the Supplementary material as suggested.

On page 13 and in the Fig. 4B caption the authors report "Chi" values. A "chi-squared" value appears to be more appropriate to describe how well a computed scattering pattern fits the observed experimental SAXS data.

It should be mentioned what program was used to calculate the theoretical SAXS profiles from the models presented in Fig. 4C.

We changed all chi values to chi-squared values.

The software is added to the Methods section (**lines 731-732**) and Suppl. Table 4.

On page 7 it is mentioned that only Ric8A residues 1-422 "could be modeled based on the electron density, suggesting that the C-terminal tail of Ric8A is flexible". A typical approach to interpreting SAXS data from partially flexible proteins would be using an ensemble of models instead of a single model,

however the fits in Fig. 4B and in Suppl. Fig. 13 appear to be from one model each - it is not clear why (the authors even cite the relevant paper Pelikan et al. 2009). Fitting each SAXS data set with an ensemble of models instead of a single model could improve the presented fits.

We have added the 3-state model from the BILBOMD modeling output and the fit of this model to experimental SAXS data in Suppl. Fig. 16 (**lines 287-291**).

A minor issue: Fig. 4A caption says "Denaturation of 9.1 μ M Ric8A1-452" but in the figure we read "Ric8A1-492".

This has been corrected.

Reviewers' comments:

Reviewer #1 (Remarks to the Author):

Overall, the authors have made a good effort and the manuscript has been improved by providing more clarity. This reviewer however feels that there are still a few questions in the way that the data have been interpreted, in particular with respect to structural modeling.

a) Despite valuable additions, the algorithm for ranking the models generated remains somewhat incomplete. Does the structural modeling algorithm - as described in Suppl Fig 13 - take into account data ambiguity (e.g. false positive cross-links)? Are there cross-links that provide more confidence than others and if so how are they quantified? In other words, can one weight the cross-linking restraints? Also the integration of MD simulations to the workflow is definitely a clever addition and should take into account protein flexibility. Both MD and cross-linking MS can capture flexibility and coexisting states. It will be interesting to see how does the fitting of the cross-linking restraints to the structure evolve over the course of the simulations? Subsequent clustering of the last MD frames (e.g. 10-20ns) may reveal further conformational variability

b) In lines 273-274, the authors state "Clustering of 212 models yielded two main clusters I and II that were comprised of 50 and 20

models, respectively". It is unclear what happened to the rest of 142 models? Perhaps the authors want to rephrase/clarify further this sentence.

c) The authors use 30Å as (euclidean) distance restraint for cross-linking. This seems not unreasonable and possibly captures the information needed. Such distance however allows only 6 Å as an extra room for flexibility and this may not be enough? I was therefore wondering whether the modelling workflow would lead to different results (and if yes how much different) if for instance a 35 Å restraints will be used.

Reviewer #2 (Remarks to the Author):

Major editorial comments:

The authors tried to address my key concern with the first submission by testing Ric8 interaction mutants with Galpha peptides, but not full length intact Galpha subunits as requested. Nor did they test individual residues they assign importance to from their SAXS and MD simulations which is essential to test conclusions from such low resolution techniques. I stand by my earlier request in the last review that they need to test their structural hypotheses using Ric8 mutants with full-length alpha subunits to see how relevant/important these are in a more physiological situation. Impact is "markedly" diminished if they elect to leave their studies at this level of analysis.

Minor editorial comments (in order of occurrence):

1) Line 74. "data at this level have not been available for Ric8 proteins," please clarify what level you are talking about.

2) Line 96 "provides a necessary foundation" delete the unnecessary word "necessary"

3) Line 124 "is of markedly higher affinity" replace with "has 40-fold higher affinity"

4) Line 142 "SmgGDG" should be "SmgGDS"

5) Line 165 "that buries 956 Å² of surface area"...is this total from both protein partners, or just Ric8? Should specify.

6)Line 155 " The Gat C-terminus is unaffected by lattice contacts"...well could always indirectly be affected. Maybe better to say that "The Gat C-terminus is not directly involved in crystal contacts"?

7) Lines 167-168. Awkward wording. "Gat335-346 form an α -helix that rests over the neutral surface of Ric8A, whereas Gat347-350 is in an extended conformation and lies over the positively charged surface". By saying "the" neutral surface and "the" positively charged surface one is implying that there are no other such surfaces on Ric8, but there certainly are.

8) 174-175. replace "pi-pi ""s with the Greek equivalent.

9) 175. "which is held in place by the T-shaped" I am sure there is a more technically correct term to use other than "T-shaped".

10) 181. "Such a scaffolding interaction could underlie the dramatic increase "...delete "dramatic". Your drama may not be my own.

11) lines 208-10. ". Notably, Ric8A was capable of binding G α 363-380, albeit with a lower affinity than to Gat333-350 (Suppl. Fig. 10A,B,C)." Cite some values here.

12) lines 258-291. This is more of the same problem observed in the original submission where the authors tend to put detailed methods in the results. Move most of this to the methods, retaining comments that summarize the key findings. The fact that they feel compelled to load this section with methods suggests that there aren't many consequential findings.

13) Line 293. "Mechanistic insights from modeling of the Ric8A/G α complex". This section is too long and again infused with material that should be in methods. It is hard to find the key results for all the technical detail. It is odd that there is far more technical detail for methods that give low resolution information, no matter how carefully implemented, than for the crystallographic data which is high resolution.

14) Lines 383-385. "Another fundamental difference between GPCRs

and Ric8A is that, whereas the former interacts with G $\alpha\beta\gamma$ without directly engaging G $\beta\gamma$, the modelled Ric8A/G α interface cannot accommodate the heterotrimer." Omit because redundant with similar passage in intro.

15) Line 388 "was found to be indispensable for stability of the protein". Um, the protein is stable enough without this region to be expressed. Just not as stable. So it is not "indispensable" as claimed. Again, please avoid adjectives that give the impression of oversell and that are technically inaccurate.

16) Lines 395-396. "which appears to displace the C-terminal tail." change to "which may displace the C-terminal tail"

17) Line 528 Specify what "X-ray detector software " is. Surely the program has a name. Also provide a reference for Scala.

18) Please report standard deviation rather than SE(M) for all data in the paper. SD is the accepted standard for publication in most journals. Yeah, SEM gives smaller error bars than SD, but that is not a good reason for using them.

19) Figure 4 lacks annotation and still needs it. A legend describing the coloring scheme would be helpful too. Overall this has improved relative to first submission, but please look globally at the figures and find ways to make them more accessible.

Reviewer #3 (Remarks to the Author):

Suppl. Table 4:

- "Source, instrument and description or reference" - only the detector model is presented, the beamline is not mentioned and the relevant paper describing the beamline is not referenced (although the beamline name is mentioned in the main text).

- "Sample temperature" - "No temp control" is not a temperature, it should be clear at what temperature the sample was measured, e.g "room temperature" or "21'C".

- "Software employed for SAXS data reduction" - it would be appropriate to reference the relevant papers: Hopkins et al. 2017 "BioXTAS RAW" and Franke et al. 2017 "ATSAS 2.8".

- "Quality-of-fit parameter (chi squared)" - It is not mentioned what models were fitted that resulted in these values. If these are just the fits of the pair distance distribution functions to the experimental data this should be clarified.

- Buffer details are not present. From the main text it is also not clear which buffer was used for SAXS studies.

The Fig. 5B caption mentioned a chi-squared value of 1.82. However page 12 of the revised manuscript says: "The range of chi² values among the 500 top energy models was 1.13-5.00. As a first step in model selection, 311 models were picked from the top 500 energy score models using the cutoff $\chi^2 < 2.0$." I.e. the presented model seems to be one of the worse fitting models. In the response to the reviewers the authors write "We have added the 3-state model from the BILBOMD modeling output and the fit of this model to experimental SAXS data in Suppl. Fig. 16" - indeed this was done for Ric8A1-492 (the chi-squared value improved up to 1.19) but it is not clear why a multi-state model was not considered for Ric8A1-452.

The authors forgot to mention the SASBDB IDs on page 29 under "Data availability" (there is just the word "SAXS" and nothing else). The document "SASBDB accession codes" mentions SASDF65 and SASDF75, however these entries are on hold and are not available for review without the shareable link which was not provided.

We have thoroughly amended our manuscript, according to the comments from the Reviewers, and have responded to each below. (Reviewers' comments are italicized.)

Reviewer #1 (Remarks to the Author):

Overall, the authors have made a good effort and the manuscript has been improved by providing more clarity. This reviewer however feels that there are still a few questions in the way that the data have been interpreted, in particular with respect to structural modeling.

a) Despite valuable additions, the algorithm for ranking the models generated remains somewhat incomplete. Does the structural modeling algorithm - as described in Suppl Fig 13 - take into account data ambiguity (e.g. false positive cross-links)? Are there cross-links that provide more confidence than others and if so how are they quantified? In other words, can one weight the cross-linking restraints? Also the integration of MD simulations to the workflow is definitely a clever addition and should take into account protein flexibility. Both MD and cross-linking MS can capture flexibility and coexisting states. It will be interesting to see how does the fitting of the cross-linking restraints to the structure evolve over the course of the simulations? Subsequent clustering of the last MD frames (e.g. 10-20ns) may reveal further conformational variability

Protein prospector ranks cross-links based on the score of a cross-linked peptide, score difference and the expectation value. These parameters are listed in the Crosslinked Peptides reports that are included as Supplemental Tables. The score of a cross-linked peptide was based on number and types of fragment ions identified, as well as the sequence and charge state of the cross-linked peptide. Only results where the score difference is greater than 0 (i.e., the cross-linked peptide match was better than a single peptide match alone) are considered. The expectation value represents how many random matches would be expected to achieve a given score or greater, in a search of a given size. Because of the score differences (19.8, 20.1 vs 6.6) and the expectation values ($3.2e-8$, $2.3e-7$ vs $7.5e-5$), the Ric8A1-492 intramolecular cross-links 408/449 and 352/449 are more robust compared to 449/375 (Suppl. Table 3). Therefore, we used them as constraints in the FloppyTail modeling of Ric8A1-452, while the third cross-link 449/375 was used in model selection. For the Ric8A1-492/miniG α crosslinking, both cross-links 122/488 and 462/21 are near the top of the cross-links list (Suppl. Table 5), although the former cross-link appears more robust. In addition, all residues with positions known from the crystal structure and identified as intramolecular or intermolecular cross-link pairs were within favorable cross-linking distances. We expanded the description of Protein Prospector scoring in the Methods section.

We calculated the distance evolution for the three crosslink pairs used in modeling/model selection during representative MD simulation of Ric8A1-452. These calculations are now added in Suppl. Figure 17C.

We have clustered 78 frames corresponding to the last 19.5 ns of one of the MD simulations of Ric8A1-452. The two largest clusters contained 33 and 22 frames. Representative frames for these clusters (identified by the Chimera software)

superimposed with RMSD of 1.5 Å over 452 CA atoms. This and the visual inspection of the two frames indicates limited conformational variability within this time frame.

b) In lines 273-274, the authors state "Clustering of 212 models yielded two main clusters I and II that were comprised of 50 and 20 models, respectively". It is unclear what happened to the rest of 142 models? Perhaps the authors want to rephrase/clarify further this sentence.

The remaining clusters were minor each containing 13 or fewer models. None of the minor clusters included models from the top 10 energy score models. Thus, these clusters totaling 142 models were excluded from further analysis. This clarification has been added to the Methods section.

c) The authors use 30Å as (euclidean) distance restraint for cross-linking. This seems not unreasonable and possibly captures the information needed. Such distance however allows only 6 Å as an extra room for flexibility and this may not be enough? I was therefore wondering whether the modelling workflow would lead to different results (and if yes how much different) if for instance a 35 Å restraints will be used.

The distance constraint of 30 Å is based on literature and it is a widely accepted value for the DSS crosslinker. The recommended range for DSS is 26-30 Å, and we are using the upper value (Merkley, E.D. et al. *Protein Sci* **23**, 747-59 (2014)). We believe that relaxing this constraint may lead to erroneous modeling.

Reviewer #2 (Remarks to the Author):

Major editorial comments:

The authors tried to address my key concern with the first submission by testing Ric8 interaction mutants with Galpha peptides, but not full length intact Galpha subunits as requested. Nor did they test individual residues they assign importance to from their SAXS and MD simulations which is essential to test conclusions from such low resolution techniques. I stand by my earlier request in the last review that they need to test their structural hypotheses using Ric8 mutants with full-length alpha subunits to see how relevant/important these are in a more physiological situation. Impact is "markedly" diminished if they elect to leave their studies at this level of analysis.

We did try to show by SEC in the first revision that the R75M mutation of Ric8A1-492 impairs its complex formation with the full-length Gat. In this revision, we strongly augment the data for the full-length Gat by employing the Avi-tagged Gat in the BLI binding assay with Ric8A1-492 and its N123E, F169R, and A173W mutants (Fig. 3G,H; Suppl. Fig. 8). Furthermore, we tested the binding Ric8A1-492 to I340Q/N343H mutant of the full-length Gα_t, and this double mutation also significantly reduced the binding (Suppl. Fig. 12). The kinetics of Ric8A1-492 and the full-length Gat (or mutants) are biphasic with a slow component that precludes achieving full saturation. We do not

know the origin for such kinetics, which could be a slow conformational change of binding partners in the complex. Nonetheless, the data suggest very large decreases in affinity for $G\alpha_t$ caused by the Ric8A mutations. Overall, these results unequivocally validate the interface seen in the crystal structure.

We are not probing the interfaces suggested by our modeling because details of interactions required for mutation design cannot be discerned with sufficient accuracy (at individual residue level) from the low-resolution data such as cross-linking.

Minor editorial comments (in order of occurrence):

1) Line 74. "data at this level have not been available for Ric8 proteins," please clarify what level you are talking about.

2) Line 96 "provides a necessary foundation" delete the unnecessary word "necessary"

3) Line 124 "is of markedly higher affinity" replace with "has 40-fold higher affinity"

4) Line 142 "SmgGDG" should be "SmgGDS"

We have made all of the appropriate corrections to the text

5) Line 165 "that buries 956 Å² of surface area"...is this total from both protein partners, or just Ric8? Should specify.

This has been specified as "total buried area divided by 2".

6)Line 155 "The Gat C-terminus is unaffected by lattice contacts"...well could always indirectly be affected. Maybe better to say that "The Gat C-terminus is not directly involved in crystal contacts"?

7) Lines 167-168. Awkward wording. "Gat335-346 form an α -helix that rests over the neutral surface of Ric8A, whereas Gat347-350 is in an extended conformation and lies over the positively charged surface". By saying "the" neutral surface and "the" positively charged surface one is implying that there are no other such surfaces on Ric8, but there certainly are.

8) 174-175. replace " π - π "s with the Greek equivalent.

We have made all of the appropriate corrections to the text

9) 175. "which is held in place by the T-shaped" I am sure there is a more technically correct term to use other than "T-shaped".

This term is commonly used in scientific literature.

10) 181. "Such a scaffolding interaction could underlie the dramatic increase "...delete "dramatic". Your drama may not be my own.

The word "dramatic" is deleted

11) lines 208-10. ". Notably, Ric8A was capable of binding Gas363-380, albeit with a lower affinity than to Gat333-350 (Suppl. Fig. 10A,B,C)." Cite some values here.

The value has been cited.

12) lines 258-291. This is more of the same problem observed in the original submission where the authors tend to put detailed methods in the results. Move most of this to the methods, retaining comments that summarize the key findings. The fact that they feel compelled to load this section with methods suggests that there aren't many consequential findings.

13) Line 293. "Mechanistic insights from modeling of the Ric8A/G α complex". This section is too long and again infused with material that should be in methods. It is hard to find the key results for all the technical detail. It is odd that there is far more technical detail for methods that give low resolution information, no matter how carefully implemented, than for the crystallographic data which is high resolution.

Significant portions with experimental details were moved from these paragraphs (lines 258-291, 293) to the Methods section as requested.

14) Lines 383-385. "Another fundamental difference between GPCRs and Ric8A is that, whereas the former interacts with G $\alpha\beta\gamma$ without directly engaging G $\beta\gamma$, the modelled Ric8A/G α interface cannot accommodate the heterotrimer." Omit because redundant with similar passage in intro.

This sentence has been shortened and rephrased.

15) Line 388 "was found to be indispensable for stability of the protein". Um, the protein is stable enough without this region to be expressed. Just not as stable. So it is not "indispensable" as claimed. Again, please avoid adjectives that give the impression of oversell and that are technically inaccurate.

"indispensable" has been replaced with "essential".

16) Lines 395-396. "which appears to displace the C-terminal tail." change to "which may displace the C-terminal tail"

This has been changed as suggested.

17) Line 528 Specify what "X-ray detector software " is. Surely the program has a name. Also provide a reference for Scala.

The XDS and Scala software has been referenced.

18) Please report standard deviation rather than SE(M) for all data in the paper. SD is the accepted standard for publication in most journals. Yeah, SEM gives smaller error bars than SD, but that is not a good reason for using them.

All SEM values and error bars have been changed to SD. In some instances, the values shown have not changed, i.e if SE is 0.05 and SD is 0.09, both values may be shown as ± 0.1

19) Figure 4 lacks annotation and still needs it. A legend describing the coloring scheme would be helpful too. Overall this has improved relative to first submission, but please look globally at the figures and find ways to make them more accessible.

Color bar depicting the coloring scheme has been added, and the bound Gat peptide has been indicated by arrow in Figure 4.

Reviewer #3 (Remarks to the Author):

Suppl. Table 4:

- "Source, instrument and description or reference" - only the detector model is presented, the beamline is not mentioned and the relevant paper describing the beamline is not referenced (although the beamline name is mentioned in the main text).

According to the Bio-CAT beamline scientist, the latest configuration of the instrument has not been published yet. A general description for the configuration is provided in the Methods section.

- "Sample temperature" - "No temp control" is not a temperature, it should be clear at what temperature the sample was measured, e.g "room temperature" or "21°C".

- "Software employed for SAXS data reduction" - it would be appropriate to reference the relevant papers: Hopkins et al. 2017 "BioXTAS RAW" and Franke et al. 2017 "ATSAS 2.8".

Suppl. Table 4 has been revised as suggested.

The references to software have been added to the Methods section.

- "Quality-of-fit parameter (chi squared)" - It is not mentioned what models were fitted that resulted in these values. If these are just the fits of the pair distance distribution functions to the experimental data this should be clarified.

We added this clarification as a footnote to Suppl. Table 4.

- Buffer details are not present. From the main text it is also not clear which buffer was used for SAXS studies.

Buffer description has been added to Methods.

The Fig. 5B caption mentioned a chi-squared value of 1.82. However page 12 of the revised manuscript says: "The range of chi2 values among the 500 top energy models was 1.13-5.00. As a first step in model selection, 311 models were picked from the top 500 energy score models using the cutoff $\chi^2 < 2.0$." i.e. the presented model seems to be one of the worse fitting models. In the response to the reviewers the

authors write "We have added the 3-state model from the BILBOMD modeling output and the fit of this model to experimental SAXS data in Suppl. Fig. 16" - indeed this was done for Ric8A1-492 (the chi-squared value improved up to 1.19) but it is not clear why a multi-state model was not considered for Ric8A1-452.

The FloppyTail model of Ric8A1-452 is an intermediate model for further building of the models of apo Ric8A1-492 and the complex with Ga. Thus, we needed a reasonable starting model, which we chose based on several criteria such as energy scores, cross-linking agreement, belonging to a larger cluster size, and fits to a SAXS profile. Because of the well-known inherent limitations of χ^2 , it is a better criterion for cut-off rather than a measure of accuracy of the model. The model that we chose had the best energy score, belonged to the largest cluster and had good agreement with cross-linking data.

The authors forgot to mention the SASBDB IDs on page 29 under "Data availability" (there is just the word "SAXS" and nothing else). The document "SASBDB accession codes" mentions SASDF65 and SASDF75, however these entries are on hold and are not available for review without the shareable link which was not provided.

SASBDB IDs have been included. Below, we provide a link to the deposited SAXS data for the reviewers.

<https://www.dropbox.com/sh/xwfmkhp2yfuljnf/AABMkLAYIk088UuZ5EvJ-eYLa?dl=0>

REVIEWERS' COMMENTS:

Reviewer #2 (Remarks to the Author):

From the response:

15) Line 388 "was found to be indispensable for stability of the protein". Um, the protein is stable enough without this region to be expressed. Just not as stable. So it is not "indispensable" as claimed. Again, please avoid adjectives that give the impression of oversell and that are technically inaccurate.

"indispensable" has been replaced with "essential".

Still not correct. It is not essential either. Instead of "essential for" I would say "contributes to". Looking throughout the manuscript, I think the authors misuse the word "essential" in every instance. A certain region or element or series of amino acids is only essential if one demonstrates a total loss of function when one perturbs them. If one sees a loss in a biophysical phenomenon, not a total loss, then one can only say that it makes a contribution to that process. To say otherwise is oversell, which has been a systematic problem with this paper. Especially a shame when the data could have spoken for itself.

Please go through all the instances of "essential" used and use a more scientifically accurate descriptor in its place.

Responses to Reviewer 2 comments (reviewer' comments are italicized)

REVIEWERS' COMMENTS:

Reviewer #2 (Remarks to the Author):

From the response:

15) Line 388 "was found to be indispensable for stability of the protein". Um, the protein is stable enough without this region to be expressed. Just not as stable. So it is not "indispensable" as claimed. Again, please avoid adjectives that give the impression of oversell and that are technically inaccurate.

"indispensable" has been replaced with "essential".

Still not correct. It is not essential either. Instead of "essential for" I would say "contributes to". Looking throughout the manuscript, I think the authors misuse the word "essential" in every instance. A certain region or element or series of amino acids is only essential if one demonstrates a total loss of function when one perturbs them. If one sees a loss in a biophysical phenomenon, not a total loss, then one can only say that it makes a contribution to that process. To say otherwise is oversell, which has been a systematic problem with this paper. Especially a shame when the data could have spoken for itself.

Please go through all the instances of "essential" used and use a more scientifically accurate descriptor in its place.

In this sentence, "essential" has been replaced with "important.

We use word "essential" only in the context of the role of the C-terminal tail in functions of Ric8A. In the absence of the C-terminal residues 427-492, the protein is nonfunctional.